# Conformal Path Reasoning: Trustworthy Knowledge Graph Question Answering via Path-Level Calibration

**Shuhang Lin** [* 1]  **Chuhao Zhou** [* 2]  **Xiao Lin** [3]  **Zihan Dong** [1]  **Kuan Lu** [2]  **Zhencan Peng** [1]  **Jie Yin** [† 4]
**Dimitris N. Metaxas** [† 1]

## Abstract

Knowledge Graph Question Answering (KGQA) offers grounded, interpretable reasoning, but existing methods often fail to provide reliable coverage guarantees over retrieved answers. While Conformal Prediction (CP) offers a principled framework for producing prediction sets with statistical guarantees, prior conformal KGQA methods suffer from two critical pitfalls: violated coverage guarantees due to invalid calibration, and weak score discriminability that yields excessively large prediction sets. We propose **Conformal Path Reasoning (CPR)**[1], a novel trustworthy KGQA framework built on two key innovations. First, query-level conformal calibration over path-level scores preserves exchangeability to ensure valid coverage guarantees. Second, we introduce the Residual Conformal Value Network (RCVNet), a lightweight module trained via PUCT-guided exploration to learn discriminative path-level nonconformity scores. Extensive experiments show that CPR significantly improves the Empirical Coverage Rate by 45% while reducing prediction set size by 52% on average over conformal baselines across benchmark datasets, highlighting its effectiveness for reliable conformal reasoning over knowledge graphs.

## 1. Introduction

Knowledge graphs (KGs) represent real-world facts as structured entity-relation triples, enabling explicit associations across heterogeneous concepts and supporting interpretable multi-hop reasoning (Ji et al., 2021; Guo et al., 2025). These structural advantages empower KGs as a cornerstone for question answering (QA), where reasoning paths can be traced directly to predicted answers (Lan et al., 2023; Liang et al., 2024). However, in high-stakes applications such as medical diagnostics or financial risk assessments, returning a single deterministic answer is often insufficient, as it fails to capture the ambiguity inherent in relational KGs nor provides a rigorous basis for reliability assessment. This gap motivates a shift toward trustworthy reasoning frameworks that produce set-valued predictions statistically guaranteed to contain the ground truth at a user-specified risk level (Li et al., 2024; Memmesheimer et al., 2026).

Conformal Prediction (CP) provides such a principled, distribution-free framework for producing prediction sets with finite-sample coverage guarantees (Vovk et al., 2005; Shafer & Vovk, 2008). Originally developed under the assumption of exchangeable—most commonly i.i.d.—calibration samples, CP has since been extended to the graph domain, including KG embeddings (Zhu et al., 2025a), node classification via graph neural networks (Huang et al., 2023), and score prediction in uncertain KGs (Zhu et al., 2025b). However, these methods are largely confined to atomic node-level assertions, validating individual triples rather than complex reasoning chains. Alternatively, approaches such as BetaE (Ren & Leskovec, 2020) and dynamic KG systems (Takahashi et al., 2025) instead model uncertainty implicitly through probabilistic embeddings or LLM-generated confidence scores, but lack rigorous statistical guarantees. UaG (Ni et al., 2025) attempts to apply conformal calibration into multi-step retrieval components in KGQA. However, bridging the gap between CP's formal statistical validity and the complexity of multi-hop reasoning paths remains an open challenge.

Extending CP to multi-hop KGQA reveals a fundamental theoretical tension. The statistical guarantees of CP are grounded on the *exchangeability* assumption—that calibration and test samples are drawn from the same distribution and remain invariant under permutation (Shafer & Vovk,

---

*Equal contribution: Shuhang Lin <shuhang.lin@rutgers.edu>, Chuhao Zhou <zhouchuhao0904@gmail.com> †Co-corresponding authors ¹Rutgers University ²Independent Researcher ³University of Illinois Urbana-Champaign ⁴The University of Sydney. Correspondence to: Jie Yin <jie.yin@sydney.edu.au>, Dimitris N. Metaxas <dnm@rutgers.edu>.

*Proceedings of the 43^{rd} International Conference on Machine Learning*, Seoul, South Korea. PMLR 306, 2026. Copyright 2026 by the author(s).

[1]Code is available at: https://github.com/betterCallSherlock/CPR.

*Figure 1.* Overview of Conformal Path Reasoning (CPR). (a) Hop-level calibration introduces sequential dependencies that violate the exchangeability assumption required for valid coverage guarantees, and weak score discriminability leads to excessively large prediction sets. (b) CPR learns discriminative path-level nonconformity scores with RCVNet from PUCT-curated training trajectories. (c) Path-level calibration produces compact prediction sets with reliable coverage guarantees.

2008). KGs, however, are inherently governed by strong *relational dependencies*: entities and relations form interconnected subgraphs, where local neighborhoods share information and multi-hop paths exhibit high-order sequential dependencies (Xiong et al., 2017; Guo et al., 2026). Such relational dependencies create two primary challenges.

- **Sequential dependency chains**: When CP is applied at the hop level, the prediction set at hop $k$ determines which entities are reachable at hop $k + 1$, creating a dependency among calibration scores that breaks the exchangeability assumption required for conformal guarantees (Barber et al., 2023). Prior attempts to reconcile CP with graph structure—including transductive CP that avoids data splitting via permutation tests (Vovk, 2013), neighborhood-aware splitting that preserves local topology (Huang et al., 2023), and degree-adaptive nonconformity scores (Gazin et al., 2024)—either incur prohibitive computational overhead or introduce distribution shifts that undermine coverage guarantees.

- **Non-discriminative scoring**: Even beyond exchangeability, the efficiency of conformal prediction sets depends on the discriminative quality of nonconformity scores (Lei et al., 2018). Scores derived solely from local entity similarity fail to capture path-level relevance in multi-hop reasoning. Consequently, the conformal mechanism must adopt overly conservative quantiles to satisfy coverage requirements, resulting in excessively large prediction sets that lack practical utility for action-oriented QA tasks.

These challenges suggest that both the calibration unit and scoring mechanism require reconsideration. First, hop-level calibration casts reasoning as sequential decisions

over shared intermediate nodes and local neighborhoods, introducing dependencies that violate exchangeability. In contrast, treating queries as independent units enables valid conformal calibration at the query level. This motivates a shift from hop-level to path-level calibration. Second, local semantic similarity is insufficient to assess the correctness of an entire reasoning trajectory. This motivates a path-level scoring mechanism that captures global reasoning coherence to better discriminate correct paths from plausible negatives.

Based on these insights, we propose **Conformal Path Reasoning (CPR)**, a novel trustworthy KGQA framework illustrated in Figure 1. CPR leverages a Residual Conformal Value Network (RCVNet) to learn discriminative path-level nonconformity scores from PUCT-curated training trajectories and performs path-level conformal calibration to produce compact prediction sets while maintaining valid coverages guarantee. Our main contributions are as follows:

- **Query-Exchangeability Conformal Prediction:** By encapsulating the multi-hop reasoning process into path-level scores aggregated per query, CPR bypasses relational dependencies and preserves exchangeability at the query level, enabling valid conformal guarantees.

- **Learnable Discriminative Path Scoring:** We propose RCVNet, a lightweight residual network trained on PUCT-curated trajectories to learn discriminative path-level scores that distinguish correct paths from plausible negatives, yielding tighter prediction sets with valid coverage.

- **Strong Empirical Improvements:** Experiments on multiple real-world benchmarks show that CPR substantially outperforms existing conformal baselines, achieving higher valid coverage with smaller prediction sets.

## 2. Related Work

### 2.1. Uncertainty Quantification in KGQA

Recent work has demonstrated the efficacy of path-based reasoning for KGQA. RoG (Luo et al., 2024) is a planning-retrieval-reasoning framework that generates relation paths as faithful plans grounded by KGs. PoG (Tan et al., 2025) extends this paradigm with dynamic multi-hop exploration and beam search pruning to handle multi-entity questions. While these methods achieve strong accuracy, they provide only point predictions without quantifying answer reliability, which is a critical limitation in high-stakes applications.

Uncertainty quantification in KGQA has received increasing attention, driven by the need for providing calibrated confidence to support decision-making. Some approaches model uncertainty implicitly. BetaE (Ren & Leskovec, 2020) embeds queries as Beta distributions for multi-hop logical reasoning, using the differential entropy as a proxy for query uncertainty without formal coverage guarantees. Takahashi et al. (2025) assign LLM-generated confidence scores to triples based on source quality and temporal consistency, enabling confidence-aware retrieval but operating at the triple level. Recent work has leveraged CP to provide statistical guarantees. For KG embeddings, Zhu et al. (2025a) design nonconformity measures that construct answer sets with provable coverage. UnKGCP (Zhu et al., 2025b) extends CP to uncertain KGs using entropy-normalized scores to yield adaptive prediction intervals. Beyond specific KG tasks, CF-GNN (Huang et al., 2023) establishes permutation invariance conditions for transductive node classification with topology-aware efficiency optimization.

More recently, UaG (Ni et al., 2025) extends CP to multi-hop KGQA by designing a stage-wise conformal pipeline across multiple components, including graph traversal and LLM-based answer evaluation. However, its hop-level calibration strategy introduces sequential dependencies that directly violate the exchangeability premise required for standard conformal guarantees. Our work shifts towards path-level calibration enabling valid conformal guarantees. When coupled with highly discriminative path-level scores learned via our proposed RCVNet, our method yields compact prediction sets while maintaining rigorous coverage.

### 2.2. Conformal Prediction in Large Language Models

CP provides a distribution-free framework for producing prediction sets with finite-sample coverage guarantees (Angelopoulos & Bates, 2023). Recent work has explored CP for uncertainty quantification in large language models (LLMs) (Geng et al., 2024). Conformal Language Modeling (CLM) extends risk control to sequence generation (Quach et al., 2024). LoFreeCP enables CP for black-box LLM APIs by estimating nonconformity scores via repeated sampling (Su et al., 2024). These methods have been adopted as baselines in uncertainty-aware NLP evaluation.

However, existing CP methods for LLMs primarily address uncertainty arising from stochastic text generation and treat reasoning as an isolated black box. They do not account for uncertainty induced by structured procedures such as multi-hop graph traversal. Our work extends CP to structured KG reasoning by defining nonconformity scores over complete reasoning paths, enabling coverage guarantees that respect the combinatorial nature of path-based retrieval.

### 2.3. MCTS and PUCT for Structured Search

Monte Carlo Tree Search (MCTS) is a widely used framework for structured decision-making that balances exploration and exploitation via upper confidence bounds (Kocsis & Szepesvári, 2006; Świechowski et al., 2022). A prominent extension is Predictor + Upper Confidence Bound applied to Trees (PUCT), popularized by AlphaGo and AlphaZero (Silver et al., 2017; 2018), which incorporates learned prior probabilities to guide search. By combining prior guidance with visit-count-based exploration bonuses, PUCT effectively collects informative search trajectories that can be distilled into parametric models (Silver et al., 2017). In this work, we employ PUCT during training to collect positive and negative trajectories for RCVNet, which provides path-level scores for conformal calibration at inference.

## 3. Preliminaries

### 3.1. Problem Formulation: KGQA as Path Reasoning

Given a query $q$ and a set of topic entities $\mathcal{E}_q \subseteq \mathcal{E}$ in a KG $\mathcal{G} = \{(h, r, t) \mid h, t \in \mathcal{E}, r \in \mathcal{R}\}$, the task of *Knowledge Graph Question Answering* (KGQA) is formulated as finding a set of valid reasoning paths over $\mathcal{G}$. A reasoning path $p = (e_0, r_1, \ldots, r_H, e_H)$ originates from a topic entity $e_0 \in E_q$ and terminates at a candidate answer $e_H \in \mathcal{E}$. Let $\mathcal{P}_q$ denote the set of all candidate paths for query $q$.

To quantify the reliability of these paths, we define a cost function $V(p)$ for each path $p \in \mathcal{P}_q$, where smaller values indicate more reliable paths. The final answer set $\mathcal{A}_q$ for query $q$ is then given by

$$\mathcal{A}_q = \{\pi(p) \mid p \in \hat{\mathcal{P}}(q)\},$$

where $\hat{\mathcal{P}}(q) \subseteq \mathcal{P}_q$ denotes the set of reasoning paths selected after conformal calibration for query $q$, and $\pi(p)$ denotes the terminal entity of path $p$.

### 3.2. Background: Conformal Prediction

Conformal Prediction (CP) is a practical framework for distribution-free uncertainty quantification (Angelopoulos & Bates, 2023). Under the assumption of data exchangeability,

CP provides finite-sample coverage guarantees, ensuring that prediction sets contain the true label at a user-specified risk level $\alpha$.

**Split Conformal Prediction.** Among existing CP variants, Split CP (Shafer & Vovk, 2008) lends itself to KGQA settings, where the scoring model is typically trained before inference, satisfying the requirement that the scoring function remains fixed prior to calibration. Specifically, given a calibration set $\mathcal{D}_{\text{cal}} = \{(x_i, y_i)\}_{i=1}^n$ and a nonconformity score function $S(x, y)$ that measures how poorly $y$ conforms to the prediction for $x$, Split CP computes a threshold

$$\tau_\alpha = \text{Quantile}\big(\{S(x_i, y_i)\}_{i=1}^n, \lceil (n+1)(1-\alpha) \rceil / n\big),$$

and constructs the prediction set $\mathcal{C}(x) = \{y : S(x, y) \leq \tau_\alpha\}$. The validity of this construction relies on *exchangeability*: a sequence $(Z_1, \ldots, Z_n, Z_{n+1})$ is exchangeable if its joint distribution is invariant to permutations. Under exchangeability between calibration and test samples, the prediction set achieves coverage $\mathbb{P}(y_{\text{test}} \in \mathcal{C}(x_{\text{test}})) \geq 1 - \alpha$.

### 3.3. The Pitfall of Hop-Level Calibration

As illustrated Figure 1(a), when applying CP to KGQA, a straightforward approach is to perform calibration at the hop level during multi-hop reasoning (Ni et al., 2025), since graph traversal naturally proceeds hop by hop (Xiong et al., 2017). However, this approach introduces sequential dependencies that violate the exchangeability assumption required for valid conformal guarantees.

Formally, let $S_k^{(i)}$ denote the hop-$k$ nonconformity score for calibration query $i$. Since hop-$k$ expansion depends on the preceding prediction set $\mathcal{C}_{k-1}^{(i)}$, we have $S_k^{(i)} = f(x_i, \mathcal{C}_{k-1}^{(i)})$, inducing a Markov dependency chain

$$S_1 \to \mathcal{C}_1 \to S_2 \to \cdots .$$

Crucially, the threshold $\tau_\alpha$ defining $\mathcal{C}_{k-1}^{(i)}$ is a function of the *entire* calibration set. Through this shared threshold, $S_k^{(i)}$ becomes statistically coupled with every other calibration query, rather than being a pointwise function of query $i$. Permuting calibration and test indices therefore changes the thresholds and reshapes every score, so the joint distribution of $\{S_k^{(i)}\}_{i=1}^{n+1}$ fails to be permutation-invariant, directly violating the exchangeability premise underlying split CP.

Consider a query *"Where did the director of Inception study?"*. If the hop-1 prediction set $\mathcal{C}_1$ excludes *Christopher Nolan*, the correct answer *UCL* becomes unreachable at hop-2, regardless of subsequent calibration thresholding.

This issue is intrinsic to hop-level calibration and cannot be addressed solely through improved scoring functions. Nevertheless, while reasoning paths within a query are correlated, individual queries remain independent samples from the same data distribution. This observation motivates query-level path calibration, which restores exchangeability at the query level, enabling valid conformal calibration.

## 4. Methodology

This section details the CPR framework for trustworthy KGQA with statistical coverage guarantees. CPR is built upon two key components: (1) **Discriminative Path Scoring**: a learning framework that leverages PUCT-guided exploration to collect informative positive-negative path pairs and trains RCVNet to learn discriminative path-level scores that distinguish correct paths from incorrect ones; and (2) **Query-Exchangeable Conformal Prediction (QE-CP):** A calibration procedure that exploits query-level exchangeability to provide valid coverage guarantees for KGQA. The path-level scores produced by the trained RCVNet serve as nonconformity scores for QE-CP, determining the final prediction set during inference.

### 4.1. Discriminative Path Scoring with RCVNet

A key challenge in incorporating CP into KGQA lies in deriving nonconformity scores with strong discriminative power. Heuristics based on local semantic matching or surface-level features often fail to distinguish correct paths from plausible but incorrect ones, resulting in overly large prediction sets. As summarized in Figure 1(b), we address this challenge through (1) a PUCT-based trajectory collector that generates high-quality training paths and relation-level priors, and (2) the **R**esidual **C**onformal **V**alue **Net**work (RCVNet) trained on these collected trajectories to produce discriminative path scores.

#### 4.1.1. PUCT-GUIDED TRAJECTORY COLLECTION

To address the lack of sufficient training paths in multi-hop KGQA, we employ PUCT (Silver et al., 2018) during training as a trajectory collector for RCVNet to construct positive-negative path pairs. PUCT offers two benefits: its exploration bonus encourages visiting under-explored relations, generating negative paths that are semantically plausible yet incorrect; meanwhile, its search process accumulates relation-level statistics that serve as structural priors.

**PUCT Exploration Policy.** Starting from a topic entity, PUCT iteratively expands paths by selecting relations that balance semantic relevance with exploration of under-visited relations. Let $p$ denote the current partial path ending at entity $e$. PUCT selects the next relation $r^*$ at each step according to the following criterion:

$$r^* = \arg\max_{r \in \mathcal{R}(p)} \left( Q(p, r) + c_{\text{puct}} \cdot P(p, r) \cdot \frac{\sqrt{N(p)}}{1 + N(p, r)} \right),$$
$$(1)$$

where $Q(p, r)$ denotes the empirical success rate of traversing relation $r$ from the current partial path $p$, $N(p, r)$ is the visit count of relation, and $N(p) = \sum_{r \in \mathcal{A}(p)} N(p, r)$ denotes the total number of visits to the partial path $p$. The term $\frac{\sqrt{N(p)}}{1+N(p,r)}$ serves as an exploration bonus that encourages visiting under-explored relations. $P(p, r)$ is a semantic prior over relations. It biases path expansions toward relations relevant to the query $\tilde{q}$ early in the search:

$$P(p, r) = \mathrm{softmax}_{r' \in \mathcal{R}(p)} \Big( s(\tilde{q}, r') + s(\tilde{q}, r_{1:t}) \Big)_r, \quad (2)$$

where $s(\cdot, \cdot)$ denotes semantic similarity (negative cosine distance), measuring query alignment with both the candidate relation $r'$ and the path traversed so far $r_{1:t}$. This prior prevents arbitrary exploration and thereby expedites the collection of informative trajectories.

**PUCT Collector Outputs.** Through the exploration process above, the PUCT-guided collector provides two complementary inputs for RCVNet:

(1) *Relation-Level Structural Priors.* For each relation $r$, we maintain a Beta-Bernoulli conjugate prior $\mathrm{Beta}(\alpha_r, \beta_r)$, initialized as $\mathrm{Beta}(1, 1)$ and updated after each rollout as:

$$(\alpha_r, \beta_r) \leftarrow \begin{cases} (\alpha_r + 1, \beta_r), & \text{if } r \text{ reaches answer,} \\ (\alpha_r, \beta_r + 1), & \text{otherwise.} \end{cases}$$

The posterior mean $\rho_r$ is calculated as:

$$\rho_r = \frac{\alpha_r}{\alpha_r + \beta_r}, \quad (3)$$

which indicates the Beta prior of relation $r$, distinguishing structurally relevant relations ($\rho_r \to 1$) from the peripheral ones ($\rho_r \to 0$). Therefore, $\rho_r$ provides a complementary structural prior when semantic similarity exhibits limited discriminative power for final path ranking.

(2) *Positive-Negative Path Pairs.* Let $p^* = (e_0, r_1, \ldots, r_H, e_H)$ denote a ground-truth reasoning path for query $q$, where $e_0$ is the topic entity and $e_H \in \mathcal{Y}_q$ is a correct answer entity. Let $\mathrm{pre}_h(p) = (e_0, r_1, \ldots, e_{h-1})$ denote the prefix of $p$ up to hop $h$. We further define $\tilde{p}^*$ as a path of the same length $H$ as $p^*$ that traverses a different sequence of relations while terminating at a correct answer entity. We construct:

- Positive paths: $\mathcal{P}^+ = \{p^*\} \cup \{\tilde{p}^*\}$, comprising ground-truth reasoning paths and other valid paths of the same length $H$ that terminate at a correct answer entity.

- Negative paths: $\mathcal{P}^- = \bigcup_{p \in \mathcal{P}^+} \bigcup_{h=1}^{H} \{\mathrm{pre}_{h-1}(p) \circ (r', e') : (r', e') \neq (r_h, e_h)\}$, consisting of paths that share the same ground-truth prefix up to hop $h - 1$ but deviate at hop $h$.

This hop-wise construction generates negative paths at intermediate decision points, preventing the model from assigning overly optimistic scores to paths that diverge early. These positive-negative path pairs serve as training trajectories for RCVNet to learn discriminative path scores.

### 4.1.2. RESIDUAL CONFORMAL VALUE NETWORK

With the training trajectories collected via PUCT, we introduce RCVNet, a neural scoring module that designed to learn discriminative path scores.

For each path $p$, RCVNet, parameterized by $\boldsymbol{\theta}$, takes a feature vector $\mathbf{x}(p)$ and a contextual path embedding vector $\mathbf{c}(p)$ as input to compute a path score:

$$V_{\boldsymbol{\theta}}(p) = \mathrm{RCVNet}_{\boldsymbol{\theta}}(\mathbf{x}(p), \mathbf{c}(p)), \quad (4)$$

where $\boldsymbol{\theta}$ denotes the learnable parameters. The input feature vector $\mathbf{x}(p) = [s(\tilde{q}, r_t), s(\tilde{q}, r_{1:t}), \rho_{r_t}] \in \mathbb{R}^3$ integrates local semantic similarity, path-level similarity and the learned relation prior $\rho_{r_t}$ via Eq. 3. The path contextual embedding vector $\mathbf{c}(p) = \mathrm{concat}(\mathbf{q}_{\mathrm{emb}}, \mathbf{r}_{\mathrm{emb}}, \mathbf{p}_{\mathrm{emb}}) \in \mathbb{R}^{3d}$ aggregates query, relation, and path embeddings. RCVNet further employs FiLM layers (Perez et al., 2018) to inject this high-dimensional contextual information into the processing of scalar features, enabling context-aware path scoring. Details of the RCVNet architecture are provided in Appendix C.

RCVNet is trained on PUCT-curated positive-negative path pairs $(p^+, p^-)$ by minimizing the pairwise ranking loss:

$$\mathcal{L} = \mathrm{Softplus}\left(V_{\boldsymbol{\theta}}(p^+) - V_{\boldsymbol{\theta}}(p^-)\right), \quad (5)$$

which encourages assigning lower scores to correct paths than to incorrect ones, i.e., $V_{\boldsymbol{\theta}}(p^+) < V_{\boldsymbol{\theta}}(p^-)$.

### 4.2. Query-Exchangeable Conformal Prediction

Based on the path scores produced by RCVNet, we now introduce Query-Exchangeable Conformal Prediction (QE-CP) that provides valid coverage guarantees for KGQA. As shown in Figure 1(c), QE-CP consists of two components: (1) TreeG, a tree-based candidate path retrieval algorithm, and (2) a query-level calibration procedure that preserves exchangeability for valid conformal guarantees.

**TreeG Retrieval.** To generate candidate paths, the TreeG algorithm is employed that leverages RCVNet scores $V_{\boldsymbol{\theta}}(p)$ alongside LLM-generated relational hints.

Given a query $q$, an LLM generates a set of relation hints $\mathcal{H}$ using fixed prompts with zero-temperature decoding to ensure determinism. TreeG expands paths hop-by-hop with branch-out size $B$ and active set size $A$, using a hint-augmented path score:

$$V'_{\boldsymbol{\theta}}(p) = V_{\boldsymbol{\theta}}(p) - \beta \cdot \sum_{r \in p} \max_{h \in \mathcal{H}} s(r, h), \quad (6)$$

where the second term rewards paths traversing relations aligned with the hints. At each step, every active path expands along the top-$B$ candidate relations and all resulting paths are globally ranked by $V'_{\boldsymbol{\theta}}(p)$. Only the top-$A$ paths are retained for the next hop expansion.

**Query-Level Calibration.**    As discussed in Section 3.3, hop-level calibration induces sequential dependencies that violate exchangeability. Instead, QE-CP treats the entire reasoning process of each query as a calibration unit. The structured sample for each query is defined as $Z_i = (q_i, \mathcal{P}_{q_i}, \mathcal{Y}_{q_i})$, where $q_i$ is a query, $\mathcal{P}_{q_i}$ is the retrieved path set, and $\mathcal{Y}_{q_i}$ denotes the ground-truth answer set for $q_i$.

The **nonconformity score** $S(q)$ is then defined as the minimum path score required to include a correct answer in the retrieved set $\mathcal{P}_q$:

$$S(q) = \begin{cases} \min_{p \in \mathcal{P}_q^*} V'_{\boldsymbol{\theta}}(p), & \text{if } \mathcal{P}_q^* \neq \emptyset, \\ +\infty, & \text{if } \mathcal{P}_q^* = \emptyset, \end{cases} \quad (7)$$

where $\mathcal{P}_q^* = \{p \in \mathcal{P}_q : \pi(p) \in \mathcal{Y}_q\}$ is the set of paths terminating at correct answer entities. Intuitively, $S(q)$ captures the difficulty of covering correct answers: lower scores indicate easier queries where correct paths rank more favorably.

The validity of CP relies on exchangeability between calibration and test samples. Let $\mathcal{D}_{\text{cal}} = \{q_i\}_{i=1}^{|\mathcal{D}_{\text{cal}}|}$ and $\mathcal{D}_{\text{test}} = \{q_j\}_{j=1}^{|\mathcal{D}_{\text{test}}|}$ denote the calibration and test query sets. Exchangeability is established through two observations. (i) **Query-level independence**: queries are i.i.d. samples drawn from a common distribution and are thus exchangeable. (ii) **Deterministic transformation**: the nonconformity score $S(q)$ is computed via TreeG retrieval, RCVNet scoring, and minimum aggregation (Eq. 7). By Lemma A.1 in Appendix A, deterministic functions preserve exchangeability, so $\{S(q_i)\}$ inherits exchangeability from $\{q_i\}$. This abstraction effectively encapsulates intra-query path dependencies, preserving exchangeability at the query level.

**Calibration and Inferene.**    Following standard split CP, we compute the conformal threshold as the $(1 - \alpha)$ quantile of the calibration scores:

$$\hat{\tau}_\alpha = \text{Quantile}_{1-\alpha}\left(\{S(q_i)\}_{i=1}^{|\mathcal{D}_{\text{cal}}|}\right). \quad (8)$$

For a test query $q$, the prediction set includes all retrieved paths with scores at or below this threshold:

$$\widehat{\mathcal{P}}_\alpha(q) = \{p \in \mathcal{P}_q : V'_{\boldsymbol{\theta}}(p) \leq \hat{\tau}_\alpha\}, \quad (9)$$

which induces the answer set $\mathcal{A}_q = \{\pi(p) : p \in \widehat{\mathcal{P}}_\alpha(q)\}$.

**Theorem 4.1** (Coverage Guarantee)**.** *For any query $q \in \mathcal{D}_{test}$, the prediction set $\widehat{\mathcal{P}}_\alpha(q)$ contains at least one correct reasoning path with probability at least $1 - \alpha$:*

$$\mathbb{P}\left(\widehat{\mathcal{P}}_\alpha(q) \cap \mathcal{P}_q^* \neq \varnothing\right) \geq 1 - \alpha. \quad (10)$$

*Consequently, the induced answer set satisfies* $\mathbb{P}(\mathcal{A}_q \cap \mathcal{Y}_q \neq \emptyset) \geq 1 - \alpha$.

The proof (Appendix A) follows from rank uniformity under exchangeability: the event $\{S(q) \leq \hat{\tau}_\alpha\}$ guarantees the existence of a correctly-terminating path in $\widehat{\mathcal{P}}_\alpha(q)$.

The training, calibration and inference procedure of CPR is summarized in Algorithm 1. The pseudocode of the TreeG retrieval algorithm is provided in Appendix B.

---

**Algorithm 1** Conformal Path Reasoning (CPR)

---

**Input:** Training set $\mathcal{D}_{\text{train}}$, Calibration set $\mathcal{D}_{\text{cal}}$, Test set $\mathcal{D}_{\text{test}}$, KG $\mathcal{G}$, Risk level $\alpha$, Maximum hop $H$
**Output:** Answer sets $\{\mathcal{A}_q\}_{q \in \mathcal{D}_{\text{test}}}$ with coverage guarantee
    **// Phase 1: RCVNet Training via PUCT Exploration**
 1: Initialize RCVNet parameters $\boldsymbol{\theta}$
 2: Initialize Beta priors $(\alpha_r, \beta_r) \leftarrow (1, 1)$ for all $r \in \mathcal{R}$
 3: **for all** $q \in \mathcal{D}_{\text{train}}$ **do**
 4:     **for** $h = 1, \ldots, H$ **do**
 5:         Select path via PUCT (Eq. 1)
 6:         Collect statistics and update Beta priors (Eq. 3)
 7:     **end for**
 8:     Generate positive paths $\mathcal{P}_q^+$ and negative paths $\mathcal{P}_q^-$
 9: **end for**
10: Optimize $\boldsymbol{\theta}$ by minimizing the loss $\mathcal{L}$ (Eq. 5)
    **// Phase 2: Calibration**
11: **for all** $q_i \in \mathcal{D}_{\text{cal}}$ **do**
12:     $\mathcal{P}_{q_i} \leftarrow \text{TREEGRETRIEVAL}(q_i, \mathcal{G}, H)$
13:     $S(q_i) \leftarrow \min_{p \in \mathcal{P}_{q_i} : \pi(p) \in Y_{q_i}} V'_{\boldsymbol{\theta}}(p)$ (Eq. 7)
14: **end for**
15: $\hat{\tau}_\alpha \leftarrow \text{Quantile}_{1-\alpha}\left(\{S(q_i)\}_{i=1}^{|\mathcal{D}_{\text{cal}}|}\right)$ (Eq. 8)
    **// Phase 3: Inference**
16: **for all** $q \in \mathcal{D}_{\text{test}}$ **do**
17:     $\mathcal{P}_q \leftarrow \text{TREEGRETRIEVAL}(q, \mathcal{G}, H)$
18:     $\mathcal{A}_q \leftarrow \{\pi(p) : p \in \mathcal{P}_q, V'_{\boldsymbol{\theta}}(p) \leq \hat{\tau}_\alpha\}$
19: **end for**
20: **return** $\{\mathcal{A}_q\}_{q \in \mathcal{D}_{\text{test}}}$

---

**Phase 1** trains RCVNet via PUCT-guided trajectory collection (Eq. 1–5). **Phase 2** calibrates the conformal threshold on $\mathcal{D}_{\text{cal}}$ (Eq. 7, 8). **Phase 3** performs inference to generate prediction sets with coverage guarantee (Theorem 4.1).

## 5. Experiments

### 5.1. Experimental Setup

**Datasets.**    We evaluate our method on **WebQSP** (Yih et al., 2016) and **ComplexWebQuestions (CWQ)** (Talmor & Berant, 2018) to assess its effectiveness in *risk-controlled set prediction* for KGQA. Following prior studies (He et al., 2021; Ni et al., 2025), we construct query-specific subgraphs on both datasets. To further evaluate the scalability, we also conduct experiments on **PathQuestion (PQ)** and **PathQuestion-Large (PQL)** (Zhou et al., 2018), two path-

based reasoning benchmarks where reasoning is conducted directly over the provided KGs without subgraph extraction. For each dataset, we derive a held-out calibration set by randomly sampling 10% from the original training set provided. Dataset statistics are summarized in Table 1.

*Table 1.* Dataset statistics and subgraph properties.

| Dataset | Train | Calibration | Test | #Nodes | #Edges |
|---------|-------|-------------|------|--------|--------|
| WebQSP | 2,543 | 283 | 1,628 | 1,378 (Avg.) | 4,180 (Avg.) |
| CWQ | 24,875 | 2,764 | 3,531 | 1,259 (Avg.) | 3,924 (Avg.) |
| PQ | 3737 | 416 | 1044 | 2,256 | 4,251 |
| PQL | 678 | 76 | 192 | 6,505 | 5,597 |

**Baseline Methods.** We compare our method against LLM conformal baselines and KG reasoning baselines under the same evaluation protocol, including: **CLM** (Quach et al., 2024), a logit-based CP method that applies the general risk-control framework directly to the output distribution of LLMs; **LoFreeCP** (Su et al., 2024): a CP variant designed to improve calibration under low-frequency events in language models; and **UaG** (Ni et al., 2025): a state-of-the-art KGQA framework that applies stage-wise CP with a Learn-Then-Test strategy to control reasoning errors across components.

**Evaluation Protocol and Metrics.** Given a user-specified risk level $\alpha \in (0, 1)$, CP constructs a prediction set with coverage guaranteed with probability at least $1 - \alpha$. We evaluate all methods across risk levels $\alpha \in \{0.3, 0.4 \ldots, 0.7, 0.8\}$ using the following metrics:

- **ECR (Empirical Coverage Rate)** measures the proportion of test cases for which the prediction set includes at least one correct answer: ECR $= \frac{1}{|\mathcal{D}_{\text{test}}|} \sum_{q \in \mathcal{D}_{\text{test}}} \mathbf{1} [\mathcal{A}_q \cap \mathcal{Y}_q]$. Higher values indicate more reliable coverage.

- **APSS (Average Prediction Set Size)** quantifies the efficiency of uncertainty quantification by measuring the average cardinality of prediction sets: APSS $= \frac{1}{|\mathcal{D}_{\text{test}}|} \sum_{q \in \mathcal{D}_{\text{test}}} |\mathcal{A}_q|$. Smaller values indicate more efficient and discriminative prediction sets.

- **Coverage Efficiency** measures how efficiently a method achieves coverage guarantees by capturing the trade-off between coverage reliability and prediction set compactness. It is defined as the **ECR/APSS ratio** (Lin et al., 2022). Higher values indicate that the method achieves adequate coverage with more compact prediction sets.

For conformal risk control prediction, the desired objective is to minimize the prediction set size while satisfying the target coverage (Angelopoulos & Bates, 2023). Accordingly, we evaluate ECR, APSS, and Coverage Efficiency achieved by different methods across various risk levels. All implementation details are provided in Appendix C.

## 5.2. Main Results

Table 2 reports the performance of all methods on WebQSP, CWQ, PQ, and PQL across various risk levels, where entries marked "-" indicate failure to satisfy the target coverage guarantee. Overall, CPR generally achieves the highest coverage rate while maintaining substantially more compact prediction sets as compared with UaG and LLM-based conformal baselines across all datasets.

On WebQSP, CPR achieves both higher coverage and smaller prediction set sizes than UaG. At a risk level of $\alpha = 0.5$, CPR achieves an ECR of 61.4% with an APSS of 7.14, whereas UaG yields a lower ECR of 51.8% despite a much larger APSS of 14.29. Moreover, CPR effectively mitigates the prediction set explosion exhibited by UaG at lower risk levels: At $\alpha = 0.3$, CPR achieves a higher ECR 76.8% compared to UaG's 72.3%, while dramatically reducing APSS from 98.81 to 17.77.

The performance gains are more pronounced on CWQ, a more challenging dataset due to its longer multi-hop reasoning chains. At $\alpha = 0.6$, CPR achieves an ECR of 50.2% with an APSS of 8.35. In contrast, even the best-performing baseline, UaG, achieves only 42.1% ECR with an excessive APSS of 120.20, and fails to satisfy coverage guarantees for $\alpha \leq 0.5$. In addition, CLM and LoFreeCP perform substantially worse, failing for $\alpha \leq 0.6$ as their ECR falls below the target level $1 - \alpha$.

CPR also exhibits strong performance on PQ and PQL, where reasoning is conducted directly over the KG provided without subgraph extraction, posing additional challenges from larger search spaces and denser graph structures. On PQ at $\alpha = 0.5$, CPR achieves an ECR of 55.1% with an APSS of 2.95, outperforming UaG, which attains an ECR of 54.6% with a comparable prediction set size. At lower risk levels, the advantage grows: at $\alpha = 0.4$, CPR achieves 61.3% ECR with an APSS of 6.80, while UaG fails to satisfy the coverage guarantee. CLM and LoFreeCP similarly fail for $\alpha \leq 0.6$. On PQL, CPR achieves 53.1% ECR with an APSS of 2.65 at $\alpha = 0.5$, compared to UaG's 52.1%. At $\alpha = 0.3$, CPR attains 71.3% ECR, whereas all baselines fail. Although UaG has a slightly smaller APSS at higher risk levels $\alpha \in \{0.7, 0.8\}$, CPR consistently maintains superior empirical coverage across all settings.

## 5.3. Ablation Study

We further conduct ablation studies to analyze the contribution of each component in CPR. Table 2 compares the full CPR model against two ablated variants: **CPR (w/o PUCT & RCVNet)**, denoted as **CPR Abl.1**, which removes both the PUCT trajectory collector and RCVNet and relies solely on TreeG retrieval with semantic value scores; and **CPR (w/o PUCT)**, denoted as **CPR Abl.2**, which retains RCVNet

*Table 2.* The comparison results on WebQSP, CWQ, PQ and PQL across various risk levels $\alpha$. "–" indicates failure to satisfy the coverage guarantee (ECR $< 1 - \alpha$). The best results are highlighted in **bold**.

| Dataset | Method | ECR (%) ↑ | | | | | | APSS ↓ | | | | | | Coverage Efficiency (%) ↑ | | | | | |
|---|---|---|---|---|---|---|---|---|---|---|---|---|---|---|---|---|---|---|---|
| | | 0.3 | 0.4 | 0.5 | 0.6 | 0.7 | 0.8 | 0.3 | 0.4 | 0.5 | 0.6 | 0.7 | 0.8 | 0.3 | 0.4 | 0.5 | 0.6 | 0.7 | 0.8 |
| WebQSP | UaG | 72.3 | 60.4 | 51.8 | 41.8 | 37.4 | 25.6 | 98.81 | 32.52 | 14.29 | 8.23 | 5.58 | 1.67 | 0.73 | 1.86 | 3.63 | 5.08 | 6.70 | 15.3 |
| | CLM | – | 60.2 | 51.0 | 40.1 | 35.6 | 24.6 | – | 46.69 | 14.37 | 5.74 | **2.78** | 1.82 | – | 1.29 | 3.55 | 6.99 | 12.8 | 13.5 |
| | LoFreeCP | – | 60.0 | 50.1 | 41.0 | 36.7 | 24.1 | – | 48.32 | 10.28 | 6.89 | 3.05 | **1.05** | – | 1.24 | 4.87 | 5.95 | 12.0 | 23.0 |
| | CPR (Abl. 1) | 76.0 | 65.5 | 57.3 | 53.4 | 49.8 | 43.2 | 22.05 | 14.84 | 10.10 | 8.26 | 3.80 | 2.16 | 3.45 | 4.41 | 5.67 | 6.46 | 13.1 | 20.0 |
| | CPR (Abl. 2) | 76.0 | 65.6 | 61.0 | 57.3 | 52.5 | 50.0 | 19.76 | **9.89** | 7.53 | 5.50 | 3.02 | 1.88 | 3.85 | 6.63 | 8.10 | 10.4 | 17.4 | 26.6 |
| | CPR | **76.8** | **67.3** | **61.4** | **58.3** | **55.1** | **50.8** | **17.77** | 10.10 | **7.14** | **5.45** | 3.01 | 1.62 | **4.32** | **6.66** | **8.60** | **10.7** | **18.3** | **31.4** |
| CWQ | UaG | – | – | – | 42.1 | 37.3 | 26.7 | – | – | – | 120.20 | 99.16 | 27.96 | – | – | – | 0.35 | 0.38 | 0.96 |
| | CLM | – | – | – | – | 32.2 | 22.5 | – | – | – | – | 10.48 | 5.33 | – | – | – | – | 3.07 | 4.22 |
| | LoFreeCP | – | – | – | – | 32.0 | 25.6 | – | – | – | – | 8.78 | 3.73 | – | – | – | – | 3.64 | 6.86 |
| | CPR (Abl. 1) | 72.2 | 65.5 | 57.8 | 50.0 | 42.2 | 27.6 | 33.43 | 27.84 | 20.50 | 14.41 | 10.96 | 4.61 | 2.16 | 2.35 | 2.82 | 3.47 | 3.85 | 5.99 |
| | CPR (Abl. 2) | 75.9 | 67.5 | 58.3 | 50.0 | 42.5 | 28.7 | 24.51 | 17.85 | 13.17 | 9.60 | 7.23 | 3.44 | 3.10 | 3.78 | 4.43 | 5.21 | 5.88 | 8.34 |
| | CPR | **76.6** | **68.1** | **58.8** | **50.2** | **42.8** | **29.9** | **23.98** | **15.28** | **11.40** | **8.35** | **6.08** | **3.39** | **3.20** | **4.46** | **5.16** | **6.01** | **7.04** | **8.82** |
| PQ | UaG | – | – | 54.6 | 41.3 | 37.5 | 33.1 | – | – | 3.79 | 1.56 | **1.24** | **1.01** | – | – | 14.4 | 26.5 | 30.2 | 32.8 |
| | CLM | – | – | – | – | 30.8 | 21.3 | – | – | – | – | 2.49 | 1.42 | – | – | – | – | 12.4 | 15.0 |
| | LoFreeCP | – | – | – | – | 31.2 | 22.4 | – | – | – | – | 1.98 | 1.21 | – | – | – | – | 15.8 | 18.5 |
| | CPR (Abl. 1) | – | 60.4 | 52.8 | 45.9 | 35.7 | 33.6 | – | 7.47 | 4.36 | 2.75 | 1.76 | 1.21 | – | 8.09 | 12.1 | 16.7 | 20.3 | 27.8 |
| | CPR (Abl. 2) | 70.1 | 60.8 | 54.2 | 46.3 | 38.5 | 36.4 | 22.5 | 6.92 | 3.11 | 1.64 | 1.32 | 1.19 | 3.12 | 8.79 | 17.4 | 28.2 | 29.2 | 30.6 |
| | CPR | **70.1** | **61.3** | **55.1** | **47.1** | **40.8** | **37.5** | **21.8** | **6.80** | **2.95** | **1.49** | 1.28 | 1.10 | **3.22** | **9.01** | **18.7** | **32.3** | **31.9** | **34.1** |
| PQL | UaG | – | – | 52.1 | 42.7 | 37.0 | 28.6 | – | – | 2.25 | 1.68 | **1.05** | **0.79** | – | – | 23.2 | 25.4 | 35.2 | 36.2 |
| | CLM | – | – | – | – | 33.3 | 27.6 | – | – | – | – | 8.61 | 6.18 | – | – | – | – | 3.88 | 4.47 |
| | LoFreeCP | – | – | – | – | 31.8 | 21.9 | – | – | – | – | 3.68 | 0.94 | – | – | – | – | 8.64 | 23.3 |
| | CPR (Abl. 1) | – | 60.6 | 51.4 | 42.1 | 35.1 | 31.9 | – | 7.25 | 3.47 | 2.46 | 1.62 | 1.12 | – | 8.36 | 14.8 | 17.1 | 21.7 | 28.5 |
| | CPR (Abl. 2) | 70.6 | 61.8 | 52.4 | 44.3 | 37.9 | 36.4 | 22.43 | 6.12 | 2.89 | 1.77 | 1.28 | 1.03 | 3.15 | 10.1 | 18.1 | 25.0 | 29.6 | 35.3 |
| | CPR | **71.3** | **62.7** | **53.1** | **45.2** | **39.0** | **38.5** | **20.85** | **5.87** | **2.65** | **1.51** | 1.09 | 0.95 | **6.57** | **10.7** | **20.0** | **29.9** | **35.8** | **40.5** |

and TreeG retrieval but replaces PUCT-guided exploration with a weaker strategy to generate training trajectories: negative paths are randomly sampled, while positive paths are obtained through semantic similarity search with fallback to ground-truth paths when no valid answers are retrieved.

On WebQSP at $\alpha = 0.5$, TreeG retrieval alone achieves an ECR of 57.3%. Adding RCVNet improves ECR to 61.0%, and incorporating PUCT-guided trajectory collection further increases ECR to 61.4%. These gains highlight the complementary roles of the three components: TreeG improves answer reachability, RCVNet enhances path score discrimination, and PUCT provides informative training trajectories that enable RCVNet to learn more discriminative path scores. Similar trends are observed on PQ and PQL: on PQL at $\alpha = 0.5$, ECR improves from 51.4% to 52.4% with RCVNet and further to 53.1% with PUCT guidance. These consistent gains demonstrate the effectiveness of each component across different graph reasoning settings.

### 5.4. Coverage Efficiency Analysis

Table 2 shows that CPR consistently achieves the highest Coverage Efficiency across all datasets and risk levels, demonstrating a superior trade-off between coverage reliability and prediction set compactness. On WebQSP at $\alpha = 0.5$, CPR achieves a Coverage Efficiency of 0.086, outperforming UaG by 140% (0.036). CLM and LoFreeCP achieve a lower efficiency of 0.035 and 0.049, respectively.

This performance gap increases significantly on CWQ: at $\alpha = 0.6$, CPR achieves 0.060, a $15\times$ improvement over UaG at 0.004. Notably, UaG fails to provide valid coverage for $\alpha \leq 0.5$ on CWQ, while CLM and LoFreeCP fail at $\alpha \leq 0.6$. On PQ and PQL, CPR also achieves the highest Coverage Efficiency in most settings. On PQ at $\alpha = 0.6$, CPR achieves a Coverage Efficiency of 0.323, compared to 0.265 for UaG. On PQL at $\alpha = 0.8$, CPR achieves the highest efficiency of 0.405 among all methods. While UaG occasionally achieves a smaller APSS at high risk levels on these datasets, its lower ECR leads to inferior Coverage Efficiency overall, confirming that CPR provides a more favorable reliability-specificity trade-off.

The ablation results further highlight the contribution of each component. At a higher risk level $\alpha = 0.8$ on WebQSP, CPR achieves a Coverage Efficiency of 0.314. Ablating PUCT reduces this score to 0.266, and further removing RCVNet causes a substantial decline to 0.200. These results indicate that PUCT-guided exploration provides high-quality training trajectories, enabling RCVNet to learn discriminative scores for producing tighter prediction sets. Overall, CPR achieves superior Coverage Efficiency while satisfying valid coverage guarantees across all datasets.

### 5.5. Scalability Analysis on Large-Scale KGs

To assess the scalability of CPR, we evaluate on vastly expanded subgraphs constructed from WebQSP, where all

query-relevant neighborhoods across each data split are aggregated into unified large-scale graphs. The statistics of the constructed large-scale subgraphs are summarized in Table 3, which substantially enlarges the search space that CPR must navigate.

*Table 3.* Statistics of the expanded WebQSP subgraphs

| Dataset Split | Number of Nodes | Number of Edges |
|---|---|---|
| Training | 964,293 | 2,731,989 |
| Calibration | 244,588 | 704,728 |
| Testing | 778,692 | 2,173,879 |

As reported in Table 4, CPR maintains valid coverage guarantees across all risk levels on expanded graphs. At $\alpha = 0.5$, CPR achieves an ECR of 55.3% with only a 2.96 increase in APSS despite a $\sim$600-fold graph expansion. Notably, CPR on expanded graphs still surpasses UaG on the original subgraphs in Coverage Efficiency across all risk levels. Although larger graphs incur higher APSS due to increased path density, Coverage Efficiency rises monotonically from 2.56% at $\alpha = 0.3$ to 12.3% at $\alpha = 0.8$, confirming that CPR scales effectively with graph complexity.

*Table 4.* Scalability evaluation of CPR on expanded WebQSP subgraphs across risk levels. UaG on the original subgraph setting is included as a reference baseline. "Original" refers to query-specific subgraphs averaging $\sim$1,378 nodes; "Expanded" refers to unified subgraphs with up to 778k nodes.

| Method | Graph Scale | $\alpha$ | | | | | |
|---|---|---|---|---|---|---|---|
| | | 0.3 | 0.4 | 0.5 | 0.6 | 0.7 | 0.8 |
| *ECR (%) ↑* | | | | | | | |
| UaG | Original ($\sim$1.3k nodes) | 72.3 | 60.4 | 51.8 | 41.8 | 37.4 | 25.6 |
| CPR | Original ($\sim$1.3k nodes) | **76.8** | **67.3** | **61.4** | **58.3** | **55.1** | **50.8** |
| CPR | Expanded (778k nodes) | 70.3 | 61.1 | 55.3 | 52.6 | 48.9 | 44.7 |
| *APSS ↓* | | | | | | | |
| UaG | Original ($\sim$1.3k nodes) | 98.81 | 32.52 | 14.29 | 8.23 | 5.58 | 1.67 |
| CPR | Original ($\sim$1.3k nodes) | **17.77** | **10.10** | **7.14** | **5.45** | **3.01** | **1.62** |
| CPR | Expanded (778k nodes) | 27.45 | 13.58 | 10.10 | 7.83 | 5.73 | 3.63 |
| *Coverage Efficiency (%) ↑* | | | | | | | |
| UaG | Original ($\sim$1.3k nodes) | 0.73 | 1.86 | 3.63 | 5.08 | 6.70 | 15.3 |
| CPR | Original ($\sim$1.3k nodes) | **4.32** | **6.66** | **8.60** | **10.7** | **18.3** | **31.4** |
| CPR | Expanded (778k nodes) | 2.56 | 4.50 | 5.48 | 6.72 | 8.53 | 12.3 |

## 5.6. TreeG Efficiency and Sensitivity Analysis

**Inference Efficiency.** TreeG performs deterministic tree expansion with bounded complexity. At each hop, it expands at most $A$ active paths over the top-$B$ relations, globally ranking the resulting $O(A \cdot B)$ candidates to retain the top-$A$. Over $H$ hops, the total complexity is $O(H \cdot A \cdot B \cdot \log(AB))$, linear in reasoning depth and quadratic in search budget. PUCT-based exploration is used only during offline training to provide high-quality trajectory supervision for RCVNet (See Appendix D for a PUCT rollout study). At inference, CPR relies solely on the efficient TreeG expansion with the trained RCVNet.

**TreeG Search Budget Analysis.** To study the effect of TreeG search budget, we vary the branch-out size $B$ and the active-set size $A$ on WebQSP as a case study. Figure 2 reports results at $\alpha = 0.5$, a moderate risk level, as TreeG retrieval is largely insensitive to the choice of $\alpha$. Overall, increasing $B$ and $A$ improves ECR but also increases APSS. Specifically, ECR grows from 0.582 to 0.629 as the search budget scales from $(4, 8)$ to $(64, 64)$, while APSS increases from 3.09 to 9.02, with diminishing marginal returns. Notably, the configuration $(16, 8)$ yields lower ECR than $(8, 8)$ despite comparable APSS, suggesting that excessive local branching without sufficient global capacity can degrade coverage. Based on this analysis, we set $(B, A) = (32, 32)$ in our experiments as it offers a favorable trade-off, achieving an ECR of 0.614 with an APSS of 6.79.

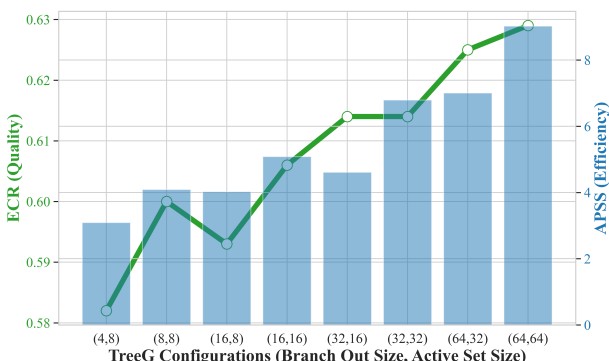

*Figure 2.* TreeG budget study on WebQSP at risk level $\alpha = 0.5$.

**Sensitivity to LLM Backbones.** To assess the sensitivity of CPR to the quality of LLM-generated relational hints, we vary the LLM backbone used for TreeG retrieval from Qwen3-8B down to Qwen3-0.6B during calibration and inference. Details are provided in Appendix E. Our results show that CPR maintains valid coverage across all backbones, while stronger LLMs mainly improve prediction set compactness. Crucially, on WebQSP at $\alpha = 0.8$, CPR without LLM-generated hints still achieves an ECR of 42.0% with an APSS of 2.54, substantially exceeding the required coverage threshold ($1 - \alpha = 20\%$) and outperforming UaG equipped with the strongest Qwen3-8B hints (25.6% ECR).

## 6. Conclusion

This paper presents CPR, a novel framework for trustworthy KGQA with statistical coverage guarantees. By performing path-level conformal calibration that leverages query exchangeability, and introducing RCVNet trained via PUCT-curated trajectories, CPR achieves valid coverage guarantees with substantially more compact prediction sets. Extensive experiments validate the effectiveness of CPR, demonstrating the potential of extending conformal prediction to complex multi-hop KGQA tasks with valid coverage guarantees.

## Acknowledgements

We sincerely thank the anonymous reviewers for their insightful comments and constructive feedback, which helped us clarify several key points, improve the presentation of our method, and strengthen the experimental evaluation. Their suggestions were invaluable in preparing the camera-ready version of this work.

## Impact Statement

This paper presents Conformal Path Reasoning (CPR), a framework for trustworthy KGQA with statistical coverage guarantees. CPR may benefit high-stakes applications such as healthcare and finance, where users need calibrated confidence in retrieved answers. We note that coverage guarantees are statistical in nature and depend on the quality of the underlying KG. In safety-critical domains, CPR should serve as a decision-support tool rather than a replacement for expert judgment. Beyond these considerations, we foresee no specific negative social consequences that must be highlighted.

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

# A. Proof of Theorem 4.1

We provide a detailed proof of the coverage guarantee for Query-Exchangeable Conformal Prediction in Section 4.2.

*Proof of Theorem 4.1.* We structure the proof in six steps.

**Step 1: Establishing Query-Level Exchangeability.** Let $q_1, \ldots, q_n$ denote the calibration queries and $q_{n+1}$ the test query. Since they are all drawn i.i.d. from the same data source, by definition of exchangeability, for any permutation $\sigma : \{1, \ldots, n+1\} \rightarrow \{1, \ldots, n+1\}$:

$$(q_1, \ldots, q_n, q_{n+1}) \stackrel{d}{=} (q_{\sigma(1)}, \ldots, q_{\sigma(n)}, q_{\sigma(n+1)}).$$

This holds because the joint distribution of i.i.d. random variables depends only on the product of marginal distributions, which is invariant to index ordering.

**Step 2: Propagating Exchangeability to Query-Path Tuples.** For a given KG $\mathcal{G}$ and a scoring function $V'_{\boldsymbol{\theta}}(p)$, define a deterministic retrieval mapping $\mathcal{R} : q \mapsto \mathcal{P}_q$. Since $\mathcal{R}$ is deterministic and the ground-truth answer set $\mathcal{Y}_q$ is also deterministically given by the data source, the tuple $(q_i, \mathcal{P}_{q_i}, \mathcal{Y}_{q_i})$ is a deterministic function of $q_i$.

We invoke the following lemma:

**Lemma A.1.** *If $(Z_1, \ldots, Z_{n+1})$ is exchangeable and $f$ is a deterministic function, then $(f(Z_1), \ldots, f(Z_{n+1}))$ is exchangeable.*

*Proof of Lemma A.1.* For any permutation $\sigma$ and measurable set $\mathcal{A}$:

$$\mathbb{P}\left((f(Z_{\sigma(1)}), \ldots, f(Z_{\sigma(n+1)})) \in \mathcal{A}\right)$$
$$= \mathbb{P}((Z_{\sigma(1)}, \ldots, Z_{\sigma(n+1)}) \in f^{-1}(\mathcal{A}))$$
$$= \mathbb{P}((Z_1, \ldots, Z_{n+1}) \in f^{-1}(\mathcal{A})) \text{ (by exchangeability of } Z_i)$$
$$= \mathbb{P}((f(Z_1), \ldots, f(Z_{n+1})) \in \mathcal{A}).$$

$\square$

Applying Lemma A.1, we conclude that $\{(q_i, \mathcal{P}_{q_i}, \mathcal{Y}_{q_i})\}_{i=1}^{n+1}$ inherits exchangeability from the queries.

**Step 3: Exchangeability of Nonconformity Scores.** Recall the nonconformity score defined in Eq. (7):

$$S(q) = \begin{cases} \min_{p \in \mathcal{P}_q^*} V'_{\boldsymbol{\theta}}(p), & \text{if } \mathcal{P}_q^* \neq \emptyset, \\ +\infty, & \text{if } \mathcal{P}_q^* = \emptyset, \end{cases}$$

where $\mathcal{P}_q^* = \{p \in \mathcal{P}_q : \pi(p) \in \mathcal{Y}_q\}$ is the set of paths terminating at correct answers.

Since $S(\cdot)$ is a deterministic function of $(q, \mathcal{P}_q, \mathcal{Y}_q)$, applying Lemma A.1 again yields that $\{S(q_1), \ldots, S(q_n), S(q_{n+1})\}$ is exchangeable.

**Step 4: Standard Split Conformal Quantile Argument.** This step constitutes the core of the proof. Let $S_i := S(q_i)$ for $i \in \{1, \ldots, n+1\}$. Following the standard split conformal framework (Angelopoulos & Bates, 2023), we define $\tau_\alpha$ as the $\lceil (n+1)(1-\alpha) \rceil$-th smallest value among the *augmented* calibration scores $\{S_1, \ldots, S_n, +\infty\}$. Formally, let $k = \lceil (n+1)(1-\alpha) \rceil$:

$$\tau_\alpha = \begin{cases} S_{(k)}, & \text{if } k \leq n, \\ +\infty, & \text{if } k > n, \end{cases}$$

where $S_{(k)}$ denotes the $k$-th order statistic of $\{S_1, \ldots, S_n\}$. The augmentation with $+\infty$ ensures the threshold is always well-defined, even when $\lceil (n+1)(1-\alpha) \rceil > n$ (which occurs only for very small $\alpha$ relative to $n$).

**Lemma A.2** (Coverage under Exchangeability). *If $(S_1, \ldots, S_n, S_{n+1})$ is exchangeable, then*

$$\mathbb{P}(S_{n+1} \leq \tau_\alpha) \geq 1 - \alpha.$$

*This holds regardless of whether ties exist among the scores.*

*Proof of Lemma A.2.* Consider the augmented sequence $(\tilde{S}_1, \ldots, \tilde{S}_{n+1}) := (S_1, \ldots, S_n, +\infty)$. By construction, $\tau_\alpha = \tilde{S}_{(k)}$ where $k = \lceil (n+1)(1-\alpha) \rceil$.

For the exchangeable sequence $(S_1, \ldots, S_{n+1})$, define the indicator $I_i = \mathbf{1}\{S_i \leq \tau_\alpha\}$ for each $i \in \{1, \ldots, n+1\}$. By exchangeability, the joint distribution is invariant under permutations, which implies:

$$\mathbb{P}(I_{n+1} = 1) = \mathbb{P}(I_i = 1) \quad \text{for all } i \in \{1, \ldots, n+1\}.$$

Let $M = \sum_{i=1}^{n+1} I_i = |\{i : S_i \leq \tau_\alpha\}|$ denote the total number of scores at or below the threshold. By construction of $\tau_\alpha$ from the augmented set, at least $k = \lceil (n+1)(1-\alpha) \rceil$ elements of $\{S_1, \ldots, S_n, +\infty\}$ satisfy $\tilde{S}_i \leq \tau_\alpha$. Since $+\infty \leq \tau_\alpha$ only when $\tau_\alpha = +\infty$, and in that case all $S_i \leq +\infty$, we have $M \geq k$ in all cases.

By symmetry:

$$\mathbb{P}(S_{n+1} \leq \tau_\alpha) = \mathbb{E}[I_{n+1}] = \frac{1}{n+1}\mathbb{E}\left[\sum_{i=1}^{n+1} I_i\right] = \frac{\mathbb{E}[M]}{n+1}.$$

Since $M \geq \lceil (n+1)(1-\alpha) \rceil$ almost surely:

$$\mathbb{P}(S_{n+1} \leq \tau_\alpha) = \frac{\mathbb{E}[M]}{n+1} \geq \frac{\lceil (n+1)(1-\alpha) \rceil}{n+1} \geq 1 - \alpha.$$

$\square$

Therefore:

$$\mathbb{P}(S(q_{n+1}) \leq \tau_\alpha) \geq 1 - \alpha. \tag{11}$$

**Step 5: From Score Coverage to Path Coverage.** We show that the event $\{S(q_{n+1}) \leq \tau_\alpha\}$ implies $\{\hat{\mathcal{C}}(q_{n+1}) \cap \mathcal{P}^*_{q_{n+1}} \neq \emptyset\}$.

When $S(q_{n+1}) \leq \tau_\alpha$ holds:

1. By definition of $S(\cdot)$, we must have $\mathcal{P}^*_{q_{n+1}} \neq \emptyset$ (otherwise $S(q_{n+1}) = +\infty > \tau_\alpha$).

2. There exists $p^* \in \mathcal{P}^*_{q_{n+1}}$ such that $V'_{\boldsymbol{\theta}}(p) = S(q_{n+1}) \leq \tau_\alpha$.

3. By the prediction set definition: $\hat{\mathcal{C}}(q) = \{p \in \mathcal{P}_q : V'_{\boldsymbol{\theta}}(p) \leq \tau_\alpha\}$, we have $p^* \in \hat{\mathcal{C}}(q_{n+1})$.

4. Since $p^* \in \mathcal{P}^*_{q_{n+1}}$ (terminates at a correct answer) and $p^* \in \hat{\mathcal{C}}(q_{n+1})$, we conclude $p^* \in \hat{\mathcal{C}}(q_{n+1}) \cap \mathcal{P}^*_{q_{n+1}}$.

Thus:

$$\{S(q_{n+1}) \leq \tau_\alpha\} \subseteq \{\hat{\mathcal{C}}(q_{n+1}) \cap \mathcal{P}^*_{q_{n+1}} \neq \emptyset\}.$$

Taking probabilities and applying Eq. (11):

$$\mathbb{P}(\hat{\mathcal{C}}(q_{n+1}) \cap \mathcal{P}^*_{q_{n+1}} \neq \emptyset) \geq \mathbb{P}(S(q_{n+1}) \leq \tau_\alpha) \geq 1 - \alpha.$$

**Step 6: Extension to Answer Entity Coverage.** From Step 5, we have $p^* \in \mathcal{P}^*_{q_{n+1}}$, which by definition means $\pi(p^*) \in \mathcal{Y}_{q_{n+1}}$. Simultaneously, $p^* \in \hat{\mathcal{C}}(q_{n+1})$ implies $\pi(p^*) \in \mathcal{A}_{q_{n+1}} = \{\pi(p) : p \in \hat{\mathcal{C}}(q_{n+1})\}$.

Therefore $\pi(p^*) \in \mathcal{A}_{q_{n+1}} \cap \mathcal{Y}_{q_{n+1}}$, yielding:

$$\mathbb{P}(\mathcal{A}_{q_{n+1}} \cap \mathcal{Y}_{q_{n+1}} \neq \emptyset) \geq 1 - \alpha.$$

This completes the proof. □

## B. Algorithm Pseudocode

Algorithm 2 details the TreeG retrieval algorithm used in both Phase 2 (calibration) and Phase 3 (inference). Given a query, TreeG first generates relational hints using an LLM and initializes paths from the topic entities. At each hop, it expands the current active paths by selecting the top-$B$ relations according to hint-adjusted scores $V'_{\boldsymbol{\theta}}(p)$ (Eq. 6), then globally ranks all candidate paths by $V'_{\boldsymbol{\theta}}(p)$ and retains the top-$A$ for expansion at the next hop.

## C. Implementation Details

**Environment.** All experiments are conducted on a cloud server running Ubuntu 22.04, equipped with one NVIDIA RTX 5090 and 25 vCPUs (Intel Xeon Platinum 8470Q).

---

**Algorithm 2** TreeG Retrieval

**Input:** Query $q$, KG $\mathcal{G}$, Maximum hop $H$, RCVNet $V_{\boldsymbol{\theta}}(\cdot)$, Active set size $A$, Branch-out size $B$
**Output:** Candidate Path Set $\mathcal{P}$
 1: $\mathcal{H} \leftarrow \text{LLM}(q)$       ▷ Generate relation hints
 2: $\mathcal{P} \leftarrow \{(e) : e \in \mathcal{E}_q\}$   ▷ Initialize with topic entities
 3: **for** $h = 1, \ldots, H$ **do**
 4:     $\mathcal{P}' \leftarrow \emptyset$
 5:     **for all** $p \in \mathcal{P}$ **do**
 6:         $\mathcal{R}_p \leftarrow$ top-$B$ relations by Eq. 6
 7:         $\mathcal{P}' \leftarrow \mathcal{P}' \cup \{p \circ (r, e') : r \in \mathcal{R}_p, (e, r, e') \in \mathcal{G}\}$
 8:     **end for**
 9:     $\mathcal{P} \leftarrow$ top-$A$ paths from $\mathcal{P}'$ by $V'_{\boldsymbol{\theta}}(p)$
10: **end for**
11: **return** $\mathcal{P}$

---

**Experimental Setup.** We evaluate CPR on WebQSP, CWQ, PQ and PQL dataset. On each dataset, the calibration set is obtained by randomly holding out 10% of the original training set. We use all-MiniLM-L6-v2, a 6-layer Transformer with approximately 22M parameters, via SentenceTransformers to encode queries, relations, and paths into 384-dimensional embeddings. RCVNet is trained using the Adam optimizer with a default learning rate of $5 \times 10^{-3}$, a batch size of 256, for 6 epochs. For inference-time TreeG retrieval, we use a default branch-out size of $B = 32$ and an active set size of $A = 32$, unless otherwise specified.

**PUCT Hyperparameters.** The exploration coefficient is set to $c_{\text{puct}} = 2$. For each query, we perform 32 rollouts with maximum depth $H$ (2 for WebQSP and PQ, 4 for CWQ and PQL). Beta priors are initialized as $\text{Beta}(1, 1)$ for all relations and updated according to Eq. 3. The Q-values $Q(p, r)$ and visit counts $N(p, r)$ are initialized to 0 at the start of each query. The semantic prior $P(p, r)$ is computed via softmax with temperature $\tau = 1.0$. A rollout is considered successful if the terminal entity belongs to the ground-truth answer set $\mathcal{Y}_q$.

**RCVNet Architecture.** As illustrated in Figure 3, RCVNet is a residual MLP that takes a 3-dimensional scalar input $\mathbf{x}(p) = [s(\tilde{q}, r_t), s(\tilde{q}, r_{1:t}), \rho_{r_t}]$ representing local similarity, path similarity, and relation prior, respectively. The network consists of 3 fully-connected layers with hidden dimension 256 and ReLU activations, followed by a linear output layer. We employ FiLM (Feature-wise Linear Modulation) conditioning: the conditioning vector $\mathbf{c}(p) = \text{concat}(\mathbf{q}_{\text{emb}}, \mathbf{r}_{\text{emb}}, \mathbf{p}_{\text{emb}}) \in \mathbb{R}^{3d}$ is passed through a 2-layer conditioner network to produce scale and shift parameters $(\gamma, \beta)$, which modulate each hidden layer via $\mathbf{h} \leftarrow \mathbf{h} \cdot (1 + \gamma) + \beta$. The output layer is initialized to zero so that the initial residual $\Delta_{\boldsymbol{\theta}}(p) \approx 0$, ensuring that the model starts from the semantic baseline.

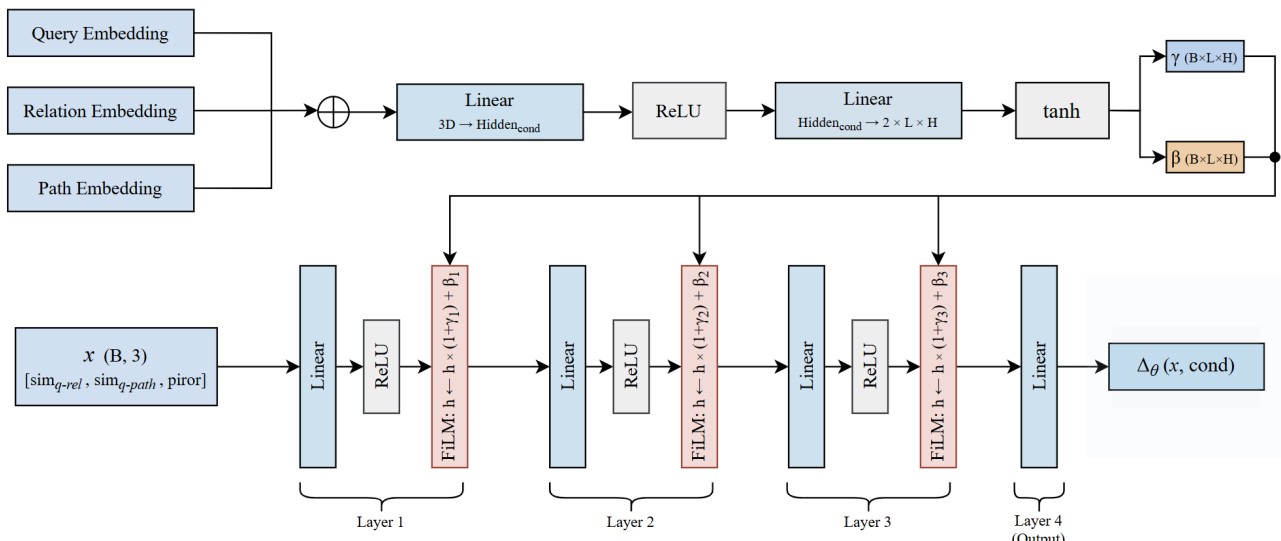

*Figure 3.* Architecture of RCVNet.

**LLM Configuration.** By default, we use Qwen3-8B served via vLLM for generating relational hints for TreeG retrieval unless otherwise specified. The model is queried with temperature 0 and thinking mode disabled to ensure deterministic outputs. For each query, the model generates up to 4 candidate relation chains with maximum hop depth $H$. We use a one-shot prompt that specifies the output format with an example:

```
You are helping a Freebase-style KGQA system.
Given a question, propose up to 4 likely
relation chains (1 to {H} hops).
Relations must be in dot-separated format like
"common.topic.image".
Output STRICT JSON ONLY in this exact format:
{"chains":[["relation1"],["relationA","relationB"]]}
Example of correct output:
{"chains":[["people.person.place_of_birth"],
["location.location.contains","people.person.nation
-ality"]]}
Question:  {question}
JSON:
```

## D. Hyperparameter Study of PUCT Rollouts

To investigate the impact of the PUCT rollout budget on CPR, we vary the number of rollouts per query on WebQSP at $\alpha = 0.7$ as a case study. As a lower-bound reference, we include a *random walk* baseline (0 rollouts), where relations are selected uniformly at random at each step rather than according to the PUCT criterion. This baseline isolates the effect of guided exploration from that of simply collecting training trajectories.

*Table 5.* Hyperparameter study on the PUCT rollout budget on WebQSP at $\alpha = 0.7$. "0 (Random Walk)" denotes trajectory collection without PUCT guidance.

| #Rollouts | ECR (%) ↑ | APSS ↓ | Coverage Efficiency (%) ↑ |
|---|---|---|---|
| 0 (Random Walk) | 52.5 | 3.02 | 17.4 |
| 4 | 52.9 | 3.02 | 17.5 |
| 8 | 53.4 | 3.02 | 17.7 |
| 16 | 54.6 | 3.01 | 18.1 |
| 32 | 55.1 | 3.01 | 18.3 |
| 64 | 56.4 | 2.98 | 18.9 |

As reported in Table 5, several consistent trends emerge. First, even a minimal budget of 4 rollouts yields a marginal but measurable improvement over random walk, increasing ECR from 52.5% to 52.9%, confirming that PUCT-guided trajectory collection provides supervision signals beyond random path sampling. Second, the performance of CPR improves steadily as the rollout budget increases, with Coverage Efficiency rising from 17.4% at 0 rollouts to 18.3% at 32 rollouts, which corresponds to an improvement of 4.57%, while APSS remains stable. This indicates that additional rollouts improve score discriminability without inflating prediction set size. Third, increasing to 64 rollouts yields further improvements (18.9%), albeit with smaller incremental gains. Based on this analysis, we adopt 32 rollouts as the default configuration throughout our experiments, as it offers a favorable trade-off between trajectory quality and computational cost.

## E. Sensitivity to LLM Backbones

We analyze the sensitivity of CPR to the quality of LLM-generated relational hints by varying the LLM backbone

used for TreeG retrieval during calibration and inference, ranging from Qwen3-8B down to Qwen3-0.6B, and further including a setting where LLM hints are entirely removed. To contextualize these results, we also report the corresponding performance of UaG under the same LLM configurations. All experiments are conducted on WebQSP at $\alpha = 0.8$, and the comparison results are presented in Table 6.

*Table 6.* Ablation study of LLM hint quality on WebQSP at $\alpha = 0.8$. "w/o LLM Hints" denotes TreeG retrieval with relation hints removed. "Failed" indicates ECR $< 1 - \alpha$, i.e., the coverage guarantee is violated.

| LLM Setting | CPR | | UaG | |
|---|---|---|---|---|
| | ECR (%) ↑ | APSS ↓ | ECR (%) ↑ | APSS ↓ |
| Qwen3-8B | **50.8** | **1.62** | 25.6 | 1.67 |
| Qwen3-4B | 45.6 | 2.45 | 22.6 | 2.70 |
| Qwen3-1.7B | 42.8 | 2.49 | 21.9 | 2.67 |
| Qwen3-0.6B | 42.3 | 2.50 | 20.4 | 2.75 |
| w/o LLM Hints | 42.0 | 2.54 | 19.8 (Failed) | 2.81 |

As shown above, CPR exhibits graceful degradation as the quality of LLM-generated hints decreases. Reducing the LLM backbone from Qwen3-8B to Qwen3-0.6B leads to a moderate increase in APSS (from 1.62 to 2.50) and a corresponding decline in ECR (from 50.8% to 42.3%), yet CPR maintains valid coverage across all configurations. Crucially, even without LLM-generated hints, CPR achieves an ECR of 42.0% with an APSS of 2.54, which remains comfortably above the required coverage level $1-\alpha = 20\%$. Moreover, its ECR remains markedly higher than that of UaG (25.6%) equipped with the strongest Qwen3-8B hints.

In contrast, UaG exhibits considerably greater reliance on the quality of LLM-generated hints. Its ECR declines more sharply as the LLM backbone shrinks, and without LLM hints, UaG fails to satisfy the coverage guarantee (ECR $= 19.8\% < 1-\alpha = 20\%$). These results reflect a fundamental difference in how the two methods utilize LLM guidance: in CPR, LLM hints act as a soft signal for hint-augmented path scoring (Eq. 6), and their absence is largely mitigated by path-level conformal calibration; in UaG, however, LLM outputs constitute a hard dependency within the stage-wise pipeline, rendering coverage guarantees significantly more vulnerable to degradation in LLM guidance.

Overall, these results demonstrate that the statistical validity of CPR is largely independent of LLM quality. While stronger LLMs provide incremental efficiency gains by yielding tighter prediction sets, the conformal coverage guarantee remains valid under degraded LLM guidance and even in the absence of LLM-generated hints.

