# OpenReview forum: "Conformal Path Reasoning: Trustworthy Knowledge Graph Question Answering via Path-Level Calibration"
_ICML.cc/2026/Conference — ICML 2026 regular_

### Official Review · Reviewer_6PSq · 2026-03-01

**Soundness:** 3
**Presentation:** 3
**Significance:** 3
**Originality:** 4
**Overall Recommendation:** 5
**Confidence:** 3

**Summary:**

This paper proposes Conformal Path Reasoning (CPR), a framework for Knowledge Graph Question Answering (KGQA) that provides reliable answer sets with rigorous statistical coverage guarantees. The authors address two critical flaws in applying Conformal Prediction to multi-hop reasoning: violated exchangeability caused by hop-level dependencies, and excessively large prediction sets caused by weak scoring functions.

**Compliance With Llm Reviewing Policy:**

Affirmed.

**Final Justification:**

All my concerns have been resolved, and I maintain a positive review.

**Key Questions For Authors:**

The TreeG retrieval process utilizes relational hints generated by Qwen3-8B to assist in expansion. If no LLM hints are used at all (relying solely on graph structure and RCVNet scores), how significantly would the system's average predicted set size (APSS) be affected?

The paper mentions that experiments were conducted on "query-specific subgraphs" to reduce computational overhead. If TreeG reasoning and PUCT trajectory collection were directly applied to an unpruned, extremely sparse global knowledge graph, what substantial changes would occur in computational complexity and coverage efficiency?

In the experimental setup, the PUCT's rollout budget is fixed at 32 per query. For complex tasks requiring longer reasoning chains (e.g., >4 hops), are these 32 rollouts still sufficient to provide an adequate number of positive-negative sample pairs for training RCVNet? Does PUCT's exploration cost increase exponentially with the maximum hop count H?

**Limitations:**

The framework relies on the Qwen3-8B model to generate zero-temperature relational hints for path expansion ; compared to the query-specific subgraphs used in the experiments , deploying the TreeG and PUCT algorithms on a full, unpruned global knowledge graph faces scalability challenges; and utilizing 32 PUCT rollouts per query forms a computational bottleneck when reasoning tasks require depths exceeding four hops. The authors should be rewarded for explicitly stating these boundary conditions, which will significantly enhance the paper's reference value for practitioners.

**Strengths And Weaknesses:**

Strengths:

The paper points out that existing KGQA conformal prediction methods (such as UaG) introduce Markovian sequential dependencies when calibrated at the "Hop" level, thereby undermining the theoretical foundation of conformal prediction. By elevating the calibration unit to the "Query" level, exchangeability is fundamentally restored, accompanied by a complete mathematical proof (Theorem 4.1 and Appendix A), forming an elegant theoretical closure.

A common issue with conformal prediction in practical applications is excessively large prediction sets. The paper goes beyond theoretical adjustments by introducing PUCT to collect high-quality positive and negative exploration trajectories, combined with residual learning via RCVNet. This design of "offline intensive exploration (PUCT) + online efficient scoring (TreeG+RCVNet)" effectively enhances score distinctiveness and significantly reduces the prediction set size.

On both the WebQSP and CWQ datasets, CPR strictly met the coverage requirements at all risk levels, whereas the baseline models for comparison (such as UaG, CLM, LoFreeCP) exhibited coverage failures at lower risk levels.

Weaknesses:

(1) During the initialization of path expansion, the TreeG retrieval algorithm relies on "relational hints" generated by zero-temperature decoding from the Qwen3-8B model. This means that CPR's actual compactness (i.e., prediction set size) heavily depends on the quality of this LLM's prior knowledge. If the hints provided by the LLM are of poor quality, although conformal mechanisms theoretically still guarantee coverage, it will lead to inflated scores for numerous useless paths, causing a sharp expansion in prediction sets. The paper lacks an ablation analysis of system robustness when LLM hint quality deteriorates.

(2) The paper explicitly mentions in the experimental setup that to reduce computational costs and ensure answer reachability, experiments were conducted on "query-specific subgraphs" rather than the complete Freebase global knowledge graph. In real-world industrial-scale KGQA systems, faced with a full knowledge graph containing hundreds of millions of nodes, TreeG's branching factor and PUCT's exploration space would grow exponentially. Whether the current search budget of (B, A) = (32, 32) remains efficient on a global graph is questionable.

(3) The paper acknowledges that PUCT incurs high computational costs due to the need for multiple random rollouts. The current hyperparameter setting performs only 32 rollouts per query. For more complex multi-hop reasoning (e.g., greater than 4 hops), is 32 rollouts sufficient to cover enough positive and negative sample pairs to train a highly discriminative RCVNet? This requires further complexity analysis.

---

> ### Author Rebuttal · Authors · 2026-03-31
>
> **W1 & Q1: System Robustness and LLM Hints**
> We clarify that the LLM (Qwen3-8B) acts as an auxiliary guide. We use cosine similarity for TreeG retrieval: as formalized in Eq.8, hallucinated relations generated by the LLM yield low similarity scores with valid relations in the KG neighborhood, thus contributing minimally to the overall score.
>
> We conducted an ablation study on LLM hints, and the results below confirm that No LLM setting only results in a slight increase in APSS. Even CPR without LLM hints achieves an ECR of 42.0%, higher than UaG’s 25.6% with the largest Qwen3-8B LLM; UaG, by contrast, fails to meet the required ECR (19.8%, marked as Failed) without LLM hints.
>
> **Ablation Study on LLM Reliance (WebQSP, $\alpha=0.8$)**
> |LLMSetting|CPR-ECR↑|CPR-APSS↓|UaG-ECR↑|UaG-APSS↓|
> |:-:|:-:|:-:|:-:|:-:|
> |Qwen3-8B|50.8|1.62|25.6|1.67|
> |Qwen3-4B|45.6|2.45|22.6|2.70|
> |Qwen3-1.7B|42.8|2.49|21.9|2.67|
> |Qwen3-0.6B|42.3|2.50|20.4|2.75|
> |w/o LLM Hints|42.0|2.54|19.8(Failed)|2.81|
>
> **W3 & Q3: PUCT Rollouts**
> We agree that PUCT rollouts involve additional computational overhead; however, this exploration process occurs **merely offline** to generate positive and negative training paths. Crucially, the online inference does not involve PUCT, maintaining high runtime efficiency for end users.
>
> We acknowledge that the number of rollouts required depends on the complexity of KGQA tasks in terms of reasoning chain steps and density of relevant graph regions. To further test our framework, we evaluated CPR on two new benchmarks: **PathQuestion(PQ)** and **PathQuestion-Large(PQL)**, which require reasoning on larger global graphs without using query-specific subgraphs.
>
> |Datasets|#Nodes|#Edges|
> |:-:|:-:|:-:|
> |PQ|2,256|4,251|
> |PQL|6,505|5,597|
>
> Our results below on PQ and PQL show that a modest budget of 32 rollouts is sufficient to cover enough positive and negative path pairs for training a discriminative RCVNet. Specifically, at α=0.8, CPR achieves an ECR of 37.5% on PQ and 38.5% on PQL, outperforming the strongest baseline UaG (33.1% on PQ and 28.6% on PQL) by a clear margin. Due to the rebuttal time constraints, we focused on these two benchmarks and will add a hyperparameter study on rollout numbers in the revision.
>
> **Results on PQ and PQL**
> |Method|Dataset|Metric|α=0.3|α=0.4|α=0.5|α=0.6|α=0.7|α=0.8|
> |:-:|:-:|:-:|:-:|:-:|:-:|:-:|:-:|:-:|
> |UaG|PQ|ECR(%)↑|–|–|54.6|41.3|37.5|33.1|
> |||APSS↓|–|–|3.79|1.56|1.24|1.01|
> |||CoverageEfficiency(%)↑|–|–|14.4|26.5|30.2|32.8|
> |CPR|PQ|ECR(%)↑|*70.1*|*61.3*|*55.1*|*47.1*|*40.8*|*37.5*|
> |||APSS↓|*21.8*|*6.80*|*2.95*|*1.49*|*1.28*|*1.10*|
> |||CoverageEfficiency(%)↑|*3.22*|*9.01*|*18.7*|*32.3*|*31.9*|*34.1*|
> |UaG|PQL|ECR(%)↑|–|–|52.1|42.7|37.0|28.6|
> |||APSS↓|–|–|*2.25*|1.68|1.05|0.79|
> |||CoverageEfficiency(%)↑|–|–|*23.2*|25.4|35.2|36.2|
> |CPR|PQL|ECR(%)↑|*71.3*|*62.7*|*53.1*|*45.2*|*39.0*|*38.5*|
> |||APSS↓|*20.85*|*5.87*|2.65|*1.51*|*1.09*|*0.95*|
> |||CoverageEfficiency(%)↑|*6.57*|*10.7*|20.0|*29.9*|*35.8*|*40.5*|
>
> “–” indicates failure to satisfy the coverage guarantee (ECR < $1 - \alpha$)
>
> **W2 & Q2: Scalability on Global vs. Subgraphs**
>
> Regarding your concerns on scalability, we would like to point out that (1) PUCT's exploration occurs exclusively during offline training (see our response to W3 & Q3); (2) TreeG's inference complexity is bounded by the maximum hop count $H$, not by the global graph size. As detailed in Section 5.5 (Line 435), TreeG expands a maximum of $A$ active paths using the top $B$ relations at each hop, yielding a total time complexity of $O(H\cdot A\cdot B\cdot\log(AB))$. As this complexity is linear to the reasoning depth $H$, CPR avoids the exponential search explosion typically in sparse global KGs. Setting a limit on $H$ is standard practice to cap computational overhead in real-world applications.
>
> Our use of query-specific subgraphs followed the established evaluation pipeline of start-of-the-art methods (e.g., UaG) for a fair and reproducible comparison. It is worth noting that this remains standard practice in KGQA, as most queries target localized regions in a broader KG$^{[2,3,4]}$.
>
> We further evaluated CPR on **PQ** and **PQL**, which require reasoning over unpruned global graphs. As shown above, CPR achieves higher ECR (37.5% vs. 33.1% on PQ and 38.5% vs. 28.6% on PQL at α=0.8) and improved coverage efficiency over UaG, while maintaining compact prediction sets with comparable APSS values. We acknowledge that while PQ and PQL represent a step towards reasoning over a full graph, they do not yet reach the scale of industrial KGs. We will leave the evaluation of CPR on industry-scale KGs to future work.
>
>
> [2] Ni et al. Towards trustworthy knowledge graph reasoning: An uncertainty aware perspective. AAAI 2025.
>
> [3] Luo et al. Reasoning on Graphs: Faithful and Interpretable Large Language Model Reasoning. ICLR 2024.
>
> [4] Jiang et al. UniKGQA: Unified Retrieval and Reasoning for Solving Multi-hop Question Answering Over Knowledge Graph. ICLR 2022.

---

> > ### Author Rebuttal · Reviewer_6PSq · 2026-04-03
> >
> > We thank the authors for the rebuttal and for adding new experiments. The ablation on LLM hints is particularly helpful and substantially reduces my concern that the method’s compactness gains mainly come from the auxiliary LLM. The additional PQ/PQL results and the clarification that PUCT rollouts are only used offline also strengthen the paper.
> >
> > I still do not think every issue is fully closed. The new scalability evidence is useful, but PQ and PQL remain far from the scale of truly large global knowledge graphs, so the practical behavior of TreeG and offline PUCT at much larger scale is still somewhat open. I also still want to see a more direct analysis of how many rollouts are needed as reasoning depth increases, since the current rebuttal shows that 32 works in a few additional settings but does not yet give a full rollout-depth sensitivity study.
> >
> > The rebuttal meaningfully improves my confidence in the work, but the scalability and rollout-budget questions remain only partially resolved. I would therefore keep my positive view while maintaining my current score.

---

> > > ### Author Response · Authors · 2026-04-08
> > >
> > > We sincerely thank the reviewer for taking the time to review our rebuttal.
> > >
> > > To address your remaining question regarding rollout sensitivity, we conducted an additional analysis on the impact of PUCT rollout budgets on WebQSP dataset.
> > > As shown below, increasing the rollout budget enhances performance. Compared to a small budget of 4 rollouts, employing 32 rollouts improves Coverage Efficiency by **4.57%**, providing an optimal balance between achieving a high, valid ECR and maintaining a compact prediction set.
> > >
> > > **Hyperparameter Study on PUCT Rollout Budget (WebQSP, $\alpha=0.7$)**
> > > |#Rollout|ECR(%)↑|APSS↓|Coverage Efficiency(%)↑|
> > > |:-:|:-:|:-:|:-:|
> > > |0 (Random Walk)|52.5|3.02|17.4|
> > > |4|52.9|3.02|17.5|
> > > |8|53.4|3.02|17.7|
> > > |16|54.6|3.01|18.1|
> > > |32|55.1|3.01|18.3|
> > > |64|56.4|2.98|18.9|
> > >
> > > Regarding your concerns on scalability, we further conducted a rigorous evaluation on vastly expanded subgraphs for the WebQSP dataset. This expansion increases the search space exponentially compared to our original setting, resulting in **964,293** nodes and **2,731,989** edges for training; **244,588** nodes and **704,728** edges for calibration; and **778,692** nodes and **2,173,879** edges for testing. Importantly, all queries within each phase are executed directly on these respective large-scale subgraphs.
> > >
> > > Due to the time constraints of the discussion phase, we conducted a targeted evaluation at the representative risk level of $\alpha=0.5$. As shown in the table below, CPR successfully maintains valid theoretical coverage with 55.3% in ECR. Compared to our original subgraph setting, the APSS only increased by 2.96. Considering that the graph's node count has expanded by nearly **600-fold**, this controlled growth in APSS is well within a reasonable range. Notably, even when operating on this vastly expanded search space, CPR still outperforms the UaG baseline evaluated on the original small query-specific subgraphs, achieving **51%** higher Coverage Efficiency. This further underscores CPR’s efficiency in maintaining compactness and its ability to effectively prune irrelevant paths even when navigating significantly larger and more complex knowledge graphs. We will include the complete results in the revised manuscript and discuss future directions accordingly.
> > >
> > > **Scalability Evaluation on Expanded WebQSP Subgraphs ($\alpha=0.5$)**
> > > | Method | Graph Scale | ECR (%) $\uparrow$ | APSS $\downarrow$ | Coverage Efficiency (%) $\uparrow$ |
> > > |:-:|:-:|:-:|:-:|:-:|
> > > |UaG|Original (Avg. ~1.3k nodes)|51.8|14.29|3.63|
> > > |CPR|Original (Avg. ~1.3k nodes)|61.4|7.14|8.60|
> > > |CPR|Expanded (778k nodes)|55.3|10.1|5.48|

---

### Official Review · Reviewer_gxZp · 2026-03-03

**Soundness:** 3
**Presentation:** 3
**Significance:** 4
**Originality:** 3
**Overall Recommendation:** 4
**Confidence:** 4

**Summary:**

This paper proposes a novel Knowledge Graph Question Answering (KGQA) framework named Conformal Path Reasoning (CPR), aiming to provide statistically guaranteed coverage for retrieved answers. The authors introduce Query-Exchangeable Conformal Prediction, which performs calibration at the path (query) level by calibrating path-level nonconformity scores. Besides, a Residual Conformal Value Network (RCVNet) is developed and trained using positive and negative path trajectories collected via a PUCT-guided exploration strategy. Experiments on the WebQSP and CWQ benchmarks demonstrate that CPR substantially reduces the Average Prediction Set Size (APSS) while maintaining valid coverage guarantees, significantly improving coverage efficiency compared to existing conformal baselines.

**Compliance With Llm Reviewing Policy:**

Affirmed.

**Final Justification:**

From the authors' rebuttal, some of the concerns have been addressed. Thus, I maintain the recommendation of this paper as weak accept.

**Key Questions For Authors:**

1. The TreeG retrieval process heavily depends on LLM-generated relation hints. If TreeG search is performed without LLM hints and relies solely on RCVNet scores, how would coverage and prediction set size degrade?
2. In longer reasoning chains or highly dense graph regions, the search space of the PUCT tree grows exponentially. Have the authors evaluated whether the current limited number of rollouts is sufficient to provide high-quality negative feedback for RCVNet under such extreme scenarios?

**Limitations:**

Yes.

**Strengths And Weaknesses:**

Strengths:
1. The paper precisely identifies a fundamental theoretical flaw in recent work that applies hop-level calibration in multi-hop reasoning, namely the violation of exchangeability.
2. The introduction of a PUCT-based trajectory collector, inspired by reinforcement learning, effectively balances exploration and exploitation through upper confidence bounds. This mechanism supplies RCVNet with high-quality positive and contrastive negative paths, enabling the model to distinguish correct paths from plausible but incorrect ones with strong discriminative capability.
3. The method demonstrates strong robustness at low risk levels. Particularly on the more complex CWQ dataset, when strong baselines such as CLM, LoFreeCP, and UaG fail to meet coverage requirements, CPR is still able to produce compact prediction sets that satisfy theoretical guarantees. This highlights both the practical and theoretical strength of the framework.

Weaknesses:
1. During inference, the TreeG retrieval algorithm relies on a frozen LLM to generate relation hints that guide path expansion. If the LLM hallucinates or fails to suggest relevant relations, correct reasoning paths may be prematurely pruned. The paper lacks a thorough analysis of such cascading errors and does not systematically evaluate the sensitivity of the framework to the quality of LLM-generated hints.

2. Although distilling PUCT into RCVNet ensures efficient online inference, the training phase requires dozens of PUCT rollouts per query. This introduces substantial offline computational cost. For extremely large-scale graphs with highly dense nodes or long reasoning chains, the scalability of the trajectory collection mechanism may become problematic.

3. The core architecture of RCVNet is primarily composed of multi-layer perceptrons (MLPs) and FiLM-based conditional modulation layers. For real-world graph data with highly complex topology and potentially multimodal semantics, such relatively simple feature interaction mechanisms may impose representational limitations, potentially constraining performance in more challenging settings.

---

> ### Author Rebuttal · Authors · 2026-03-31
>
> **W1 & Q1: LLM Reliance and Hallucinations**
>
> Thank you for regarding the potential for cascading errors if the LLM hallucinates relation hints. To mitigate this, our framework treats the LLM (Qwen3-8B) as a soft hint guide.
>
> Specifically, we use cosine similarity for TreeG retrieval. As formalized in Eq.8, hallucinated relations generated by the LLM yield low similarity scores with valid relations in the KG neighborhood, thus contributing minimally to the overall score.
>
> To answer your question about performance without LLM hints, we conducted an additional ablation study. As shown in the table below, removing the LLM or using smaller LLMs results in a slight increase in Average Prediction Set Size (APSS), but crucially, the Empirical Coverage Rate (ECR) remain valid. Notably, even CPR without LLM hints achieves an ECR of 42.0%, higher than UaG’s 25.6% with the largest Qwen3-8B LLM; UaG, by contrast, fails to meet the required ECR (19.8%, marked as "Failed") without LLM hints.
>
> **Ablation Study on LLM Hint Quality (WebQSP, $\alpha=0.8$)**
> |LLMSetting|CPR-ECR↑|CPR-APSS↓|UaG-ECR↑|UaG-APSS↓|
> |:-:|:-:|:-:|:-:|:-:|
> |Qwen3-8B|50.8|1.62|25.6|1.67|
> |Qwen3-4B|45.6|2.45|22.6|2.70|
> |Qwen3-1.7B|42.8|2.49|21.9|2.67|
> |Qwen3-0.6B|42.3|2.50|20.4|2.75|
> |w/o LLM Hints|42.0|2.54|19.8(Failed)|2.81|
>
> **W2 & Q2: PUCT Scalability and Rollouts**
>
> We agree that PUCT introduces additional computational overhead during the rollout phase. However, we intentionally designed CPR so that this exploration occurs **exclusively during the offline training phase**.
>
> The critical bottleneck for real-world scalability is typically the online calibration and testing (inference) phase. Because we distill the expensive PUCT knowledge into RCVNet offline, the online inference phase requires zero PUCT rollouts and remains scalable.
>
> We acknowledge that the number of rollouts required depends on the complexity of KGQA tasks subject to reasong chain steps and density of relevant graph regions. Even on the larger graphs of PathQuestion(PQ) and PathQuestion-Large(PQL), RCVNet can still obtain discriminative scores with a limited number of PUCT rollouts. To empirically validate this, we evaluated CPR on these two new KGQA benchmarks, where experiments were run directly on the larger, full-size graphs provided by the datasets, without extracting query-specific subgraphs. Detailed graph statistics are provided below.
>
> **Datasets Comparison**
> |Datasets|#Nodes|#Edges|
> |:-:|:-:|:-:|
> |WebQSP|1,378 (Avg.)|4,180 (Avg.)|
> |CWQ|1,259 (Avg.)|3,924 (Avg.)|
> |PQ|2,256|4,251|
> |PQL|6,505|5,597|
>
> As shown in the results below, a budget of 32 rollouts per query is still sufficient to provide high-quality feedback: on both PQ and PQL, CPR achieves higher ECR (e.g., 37.5% vs. 33.1% on PQ and 38.5% vs. 28.6% on PQL at α=0.8) and improved coverage efficiency over UaG, while maintaining compact prediction sets with comparable APSS values.
>
> **New Results on PQ and PQL**
> |Method|Dataset|Metric|α=0.3|α=0.4|α=0.5|α=0.6|α=0.7|α=0.8|
> |:-:|:-:|:-:|:-:|:-:|:-:|:-:|:-:|:-:|
> |UaG|PQ|ECR(%)↑|–|–|54.6|41.3|37.5|33.1|
> |||APSS↓|–|–|3.79|1.56|1.24|1.01|
> |||CoverageEfficiency(%)↑|–|–|14.4|26.5|30.2|32.8|
> |CPR|PQ|ECR(%)↑|*70.1*|*61.3*|*55.1*|*47.1*|*40.8*|*37.5*|
> |||APSS↓|*21.8*|*6.80*|*2.95*|*1.49*|*1.28*|*1.10*|
> |||CoverageEfficiency(%)↑|*3.22*|*9.01*|*18.7*|*32.3*|*31.9*|*34.1*|
> |UaG|PQL|ECR(%)↑|–|–|52.1|42.7|37.0|28.6|
> |||APSS↓|–|–|*2.25*|1.68|1.05|0.79|
> |||CoverageEfficiency(%)↑|–|–|*23.2*|25.4|35.2|36.2|
> |CPR|PQL|ECR(%)↑|*71.3*|*62.7*|*53.1*|*45.2*|*39.0*|*38.5*|
> |||APSS↓|*20.85*|*5.87*|2.65|*1.51*|*1.09*|*0.95*|
> |||CoverageEfficiency(%)↑|*6.57*|*10.7*|20.0|*29.9*|*35.8*|*40.5*|
>
> “–” indicates failure to satisfy the coverage guarantee (ECR < $1 - \alpha$)
>
> **W3: RCVNet Architecture and Representational Limits**
>
> While RCVNet’s core architecture utilizes lightweight MLPs and FiLM layers, we respectfully clarify that its *inputs* are highly expressive. The network processes high-dimensional embeddings for both query and explored relation paths(Line 223 right).
>
> The scalar features mentioned in the paper represent explicitly computed structural priors and semantic similarities between these high-dimensional embeddings. By combining dense latent representations with scalar topological features, the MLPs are sufficient to capture complex semantics. This design represents a trade-off: it ensures sufficient representational capacity for real-world complexity while maintaining the online inference efficiency required for practical KGQA systems.

---

### Official Review · Reviewer_bShe · 2026-03-12

**Soundness:** 2
**Presentation:** 3
**Significance:** 3
**Originality:** 2
**Overall Recommendation:** 4
**Confidence:** 4

**Summary:**

The paper introduces Conformal Path Reasoning (CPR), a framework designed to address the limitations of existing Knowledge Graph Reasoning and Link prediction (KGRL) methods in KGQA scenarios where multiple valid answers may exist. CPR reframes KGQA as a set-valued prediction problem and aims to provide statistical coverage guarantees for its predictions. The approach combines two main components: (1) a Discriminative Path Scoring Mechanism that uses Predictor + Upper Confidence Bound applied to Trees (PUCT) to efficiently explore the knowledge graph and collect both positive and negative path pairs, with a novel RCVNet network to score and distinguish between them; and (2) Query-Exchangeable Conformal Prediction (QE-CP), which calibrates predictions at the query level by leveraging the exchangeability of queries, rather than individual hops, to ensure valid statistical guarantees. The method is evaluated on standard multi-hop KGQA benchmarks, demonstrating improved set prediction quality and coverage compared to existing KGRL and conformal prediction baselines.

**Compliance With Llm Reviewing Policy:**

Affirmed.

**Final Justification:**

I am keeping my score based on the discussion.

**Key Questions For Authors:**

* Q1- In Equation (1), Aq appears to denote terminal entities, while in Equation (3) it seems to refer to a set of relations. Could you clarify the intended meaning and resolve any notational inconsistency?
* Q2-Are the relation-level structural posterior means in RCVNet’s feature vector computed per query or are they global statistics?
* Q3- Are negative paths only sampled at the last-but-one PUCT iteration, or are intermediate negatives included? Additionally, does the LLM generate full relation chains or individual relations for hop-by-hop expansion, and is the same set reused across hops?
* Q4- How does the Beta posterior interact with semantic similarity features in RCVNet’s feature vector?
* Q5- In Figure 3 of the UaG paper, ECR and APSS values for certain risk levels (0.3, 0.4, 0.5) are reported, but are missing in the CPR submission due to coverage guarantee failures. Could you explain this discrepancy ?

**Limitations:**

No, the limitations are not adequately discussed. While the paper states there are no negative societal impacts, it omits a dedicated discussion of technical limitations. The following could be discussed in a revised manuscript:
* Early-hop calibration
* Potential overfitting of RCVNet to last-hop negatives.
* Reliance on low-dimensional features in RCVNet.

**Strengths And Weaknesses:**

Strengths:
* The CPR framework is technically coherent.
* The shift from hop-level to query-level calibration is well-motivated and theoretically justified via the exchangeability assumption.

Weaknesses:
* w1- Overfitting to Last-Hop Negatives: RCVNet is trained only on negative paths sampled at the last-but-one iteration, which may cause it to assign overly optimistic scores to paths that diverge earlier in the reasoning process. An ablation study comparing last-hop-only versus multi-hop negative sampling would clarify the impact and improve model robustness.
* w2- Low-Dimensional Feature Representation: RCVNet currently uses simple scalar features, which may be insufficient for capturing the complexity of multi-hop reasoning.

---

> ### Author Rebuttal · Authors · 2026-03-31
>
> **W1 & Q3: Negative Sampling Strategy**
>
> We would like to clarify that our RCVNet is **NOT** trained only on negative paths sampled at the last one hop. Instead, our PUCT-guided trajectory collection includes intermediate negatives: At *every* training decision point (hop $i$), we sample negative relations to replace the true relation at that specific hop $i$. The prefix (hops $1$ to $i-1$) remains identical to the ground truth, and we do not append a suffix. This hop-by-hop negative sampling strategy is specifically designed to prevent the model from assigning optimistic scores to paths that diverge early.
>
> We apologize for causing the confusion - This might stem from a notation error in line 256 left column, where inadvertently used the maximum depth $H$ instead of the intermediate step index $h$. We will rectify this in the revision.
>
> For the effectiveness of our PUCT-guided strategy in generating negative paths, we respectfully direct the reviewer to the ablation study in Section 5.3, where we compare the full framework against **CPR Abl.1** (line 354 right) that degrades the PUCT trajectory collection to generate randomly sampled negative paths. As shown in Table 2 (Page 8), this ablation undermines the model's discriminative ability, resulting in a **2.69** and **7.21** increase of APSS in WebQSP and CWQ, respectively.
>
> **W2: RCVNet Feature Dimensionality**
>
> We clarify that RCVNet does not rely on scalar features, but instead utilizes FiLM layers to modulate the scoring process with a $3d$-dimensional vector $\mathbf{c}(p)$ (line 222 right and line 642 right) concatenated with high-dimensional query, relation, and path embeddings. By combining these dense semantic representations with structural interactions, RCVNet captures the complex semantics of multi-hop reasoning while maintaining computational efficiency.
>
> **Q1: Notation Inconsistency regarding $\mathcal{A}$**
> We thank the reviewer for identifying this notation conflict. We clarify that $\mathcal{A}_q$ in Eq (1) denotes the set of predicted answer entities, while $\mathcal{A}(p)$ in Eq (3) denotes the set of available relations (actions) during path expansion. In the revision, we will rectify $\mathcal{A}(p)$ to **$\mathcal{R}(p)$** in Eq (3) and (4) to represent the set of candidate relations reachable from the current path $p$.
>
> **Q2: Structural Posterior Means**
>
> The relation-level structural posterior means are **global statistics** accumulated across all training queries , not computed per query. By calculating these across the KG, the model effectively accumulates broad structural experience and general graph connectivity priors.
>
> **Q3 (Part 2): LLM Prompting and Hop-by-Hop Expansion**
>
> During the TreeG retrieval phase, the LLM is invoked only once per query to produce candidate relation chains as relational hints (Eq. 8). These chains are then decomposed into individual relation embeddings and used across all hops to compute semantic similarities and guide hop-by-hop path expansion.
>
> **Q4: Interaction of Beta Posterior and Semantic Similarity**
>
> There is no specific interaction between the Beta posterior and the semantic similarity features. Within RCVNet, they serve as distinct structural and semantic channels within the same vector. As briefly noted in Line 273 (left column), these features are simply concatenated. RCVNet naturally learns to weight and interact these distinct signals during the training process to produce the final discriminative score.
>
> **Q5: Discrepancy in UaG Baseline Results**
>
> We sincerely thank the reviewer for this observation. The discrepancy in ECR and APSS for UaG at lower risk levels ($\alpha \in \{0.3, 0.4, 0.5\}$) arises from slight difference in implementation details. Since UaG’s original subgraph extraction pipeline is not publicly released, we followed the method from [1] to extract subgraphs, and evaluate CPR and other baselines. As shown in the table below, our subgraphs maintain average node counts similar to the UaG setup, while edge count is significantly higher. In this denser, more challenging setting, the calibration of UaG fails due to compounding errors across hops, especially at lower risk levels.
>
> **Subgraph Size Comparison**
>
> |Datasets|Avg. #Nodes|Avg. #Edges|
> |:-:|:-:|:-:|
> |WebQSP (Ours)|**1,378**|**4,180**|
> |WebQSP (UaG)|1,374|2,909|
> |CWQ (Ours)|**1,259**|**3,924**|
> |CWQ (UaG)|1,256|2,615|
>
> [1] He et al. Improving Multi-hop Knowledge Base Question Answering by Learning Intermediate Supervision Signals. WSDM 2021.

---

> > ### Author Rebuttal · Reviewer_bShe · 2026-04-06
> >
> > Thank you for your previous response. However, I remain confused by your claim that the semantic similarities and relation priors are “not scalars.” In the manuscript (line 274), you explicitly define the RCVNet input feature vector as
> >
> > $x(p)=[s(q,r_t),  s(q,r_{1:t}),  ρ_{r_t}] \in R^3$
> >
> > which clearly indicates that each of these three components is a real‑valued scalar. Could you clarify why you stated that these quantities are not scalars, given that the paper formally represents them as scalar features?
> > Additionally, since these three scalar values form only a 3‑dimensional vector, I can’t apprehend how they can meaningfully influence the output of RCVNet when the context vector c(p) is a high‑dimensional vector in $R^{3d}$. Could you elaborate on how such a low‑dimensional scalar feature vector interacts with the much higher‑dimensional contextual embedding within the FiLM‑conditioned residual network?

---

> > > ### Author Response · Authors · 2026-04-07
> > >
> > > Sorry for the confusion, and thanks for pointing out our oversight. We will change line 274 from "input vector $\mathbf{x}(p)$" to "the scalar input feature vector $\mathbf{x}(p)$ and the context embedding input vector $\mathbf{c}(p)$" to better align with L267–L268 (left column) and Eq. (6).
> > >
> > > More specifically, RCVNet takes two heterogeneous types of inputs: (1) scalar features $\mathbf{x}(p)\in\mathbb{R}^{3}$, and (2) an embedding representation $\mathbf{c}(p)\in\mathbb{R}^{3d}$, which serves as a context conditioning signal.
> > >
> > > To remain lightweight and overcome the dimensional imbalance between the two types of features, we adopt FiLM as our backbone. Specifically, FiLM modulates the features derived from $\mathbf{x}(p)$ using $\mathbf{c}(p)$ as the conditioning signal:
> > >
> > > $$\mathbf{h} \leftarrow \mathbf{h} \times (1+\boldsymbol{\gamma}(\mathbf{c}(p))) + \boldsymbol{\beta}(\mathbf{c}(p)),$$
> > >
> > > where $\mathbf{h} = f_x(\mathbf{x}(p))$ is the hidden representation computed from the scalar input, and $\boldsymbol{\gamma}(\mathbf{c}(p))$, $\boldsymbol{\beta}(\mathbf{c}(p))$ are learned scaling and shifting parameters derived from the context embedding input. In this design, the scalar input $\mathbf{x}(p)$ provides a preliminary estimate of the score, and the high-dimensional embedding $\mathbf{c}(p)$ further supplies fine-grained contextual cues through $\boldsymbol{\gamma}$ and $\boldsymbol{\beta}$, yielding discriminative scoring.

---

### Official Review · Reviewer_vo6x · 2026-03-13

**Soundness:** 3
**Presentation:** 2
**Significance:** 3
**Originality:** 3
**Overall Recommendation:** 4
**Confidence:** 2

**Summary:**

This paper studies trustworthy KGQA through query-level conformal prediction over reasoning paths. The problem is meaningful and the overall direction is reasonable. However, the paper does not communicate its main idea and contribution clearly enough. In its current form, the method feels under-motivated and the empirical validation is too limited to fully support the broader claims.

**Compliance With Llm Reviewing Policy:**

Affirmed.

**Final Justification:**

The rebuttal has addressed my main concerns.

**Key Questions For Authors:**

1. What is the main contribution the authors want readers to take away: the query-level conformal formulation, the learned path scoring, or the full combined system?

2. Why is this particular combination of components the right solution for the core KGQA problem being targeted?

3. How much of the performance gain comes from conformal calibration itself, versus better retrieval or path scoring?

4. Can the method generalize to newer or more challenging KGQA benchmarks?

**Limitations:**

There is no such a section. For example,
- the theoretical guarantee is narrower than the broader framing of trustworthy KGQA;
- it mainly provides coverage-style assurance rather than full reasoning correctness.
- the empirical evidence is too limited to support strong claims about general applicability.

**Strengths And Weaknesses:**

Strength:

- This paper addresses the calibration and reliability in multi-hop KGQA.
- The query-level conformal formulation is a sensible direction for KGQA.
- The method appears technically coherent and aims to provide a meaningful coverage guarantee.


Weakness

1. The writing and organization are weak, making it hard to understand the paper’s main message. For example,
   - The introduction moves across KG, CP, and KGQA without clearly setting up the proposed method.
   - Key concepts are introduced too late or not integrated well into the narrative; the paper would benefit from clearer definitions and better exposition.
   - Figure 1 is not effectively used to guide the paper structure.

2. The contribution is not positioned sharply enough, so the method can look like a combination of several existing components rather than a clearly justified solution.

3. The experiments are limited, with only two relatively old benchmarks (WebQSP, CWQ) and fairly light analysis.

---

> ### Author Rebuttal · Authors · 2026-03-31
>
> **Q1 & Q2 & W2: Contributions and Component Justification**
>
> The novelty of our work is a principled framework for multi-hop KGQA that provides reliable answer sets with statistical coverage guarantees. Our contribution is not merely a combination of existing components, but a unified solution that integrates theoretical rigor with a scaffolded methodological design to address the identified gap:
>
> 1. **Theoretical contribution:** We identify a critical flaw in prior work: hop-level calibration introduces sequential dependencies, violating the exchangeability assumption required by CP, as acknowledged by Reviewer 6PSq. To address this, we propose a novel **query-level conformal formulation** that satisfies exchangeability and provides valid coverage guarantee (Theorem 4.1).
> 2. **Methodological contribution:** Each component is functionally required to satisfy theoretical conditions and ensure practical utility.
>    * **PUCT** collects positive-negative path pairs via balanced exploration and exploitation, providing informative training signals for RCVNet.
>    * **RCVNet** learns discriminative path scores that separate correct paths from plausible negatives, which is key to tightening prediction sets while maintaining coverage guarantee.
>    * **TreeG** leverages RCVNet scores to prune the search space and enables efficient inference, ensuring the practical utility of our theoretically sound CP framework.
>
> The three components are seamlessly integrated into CPR; the effectiveness of each component is empirically supported through Ablation Study (see Table 2).
>
> **Q3: Performance Gains from Conformal Calibration vs. Retrieval and Path Scoring**
>
> We would like to clarify the distinct roles of these mechanisms in CPR.
>
> * **Conformal Calibration** provides a statistical guarantee that the prediction set covers the true answer at risk level $\alpha$, measured by coverage validity (ECR). This contributes to a **9.5%** average ECR improvement over the strongest baseline (UaG) across WebQSP and CWQ, while UaG struggles to maintain valid coverage at lower risk levels.
> * **Retrieval and Path Scoring (RCVNet & TreeG)** allows to satisfy the coverage guarantee while returning a smaller prediction set, measured by Average Prediction Set Size (APSS). The two components contribute to a **5.99%** average reduction in APSS across WebQSP and CWQ.
>
> **W3 & Q4: More Challenging Datasets**
>
> We further evaluated CPR on two more challenging KGQA benchmarks: **PathQuestion(PQ)** and **PathQuestion-Large(PQL)**, which require reasoning over larger global graphs without using query-specific subgraphs. Dataset statistics are presented below.
>
> **Datasets Statistics**
> |Datasets|#Nodes|#Edges|
> |:-:|:-:|:-:|
> |WebQSP|1,378 (Avg.)|4,180 (Avg.)|
> |CWQ|1,259 (Avg.)|3,924 (Avg.)|
> |PQ|2,256|4,251|
> |PQL|6,505|5,597|
>
> The results below show that CPR successfully maintains valid coverage while keeping prediction sets compact on PQ and PQL, consistent with our findings on WebQSP and CWQ. Compared to the strongest baseline (UaG), CPR improves Coverage Efficiency by 3.28% on PQ and 1.55% on PQL.
>
> **New Results on PQ and PQL**
> |Method|Dataset|Metric|α=0.3|α=0.4|α=0.5|α=0.6|α=0.7|α=0.8|
> |:-:|:-:|:-:|:-:|:-:|:-:|:-:|:-:|:-:|
> |UaG|PQ|ECR(%)↑|–|–|54.6|41.3|37.5|33.1|
> |||APSS↓|–|–|3.79|1.56|1.24|1.01|
> |||CoverageEfficiency(%)↑|–|–|14.4|26.5|30.2|32.8|
> |CPR|PQ|ECR(%)↑|*70.1*|*61.3*|*55.1*|*47.1*|*40.8*|*37.5*|
> |||APSS↓|*21.8*|*6.80*|*2.95*|*1.49*|*1.28*|*1.10*|
> |||CoverageEfficiency(%)↑|*3.22*|*9.01*|*18.7*|*32.3*|*31.9*|*34.1*|
> |UaG|PQL|ECR(%)↑|–|–|52.1|42.7|37.0|28.6|
> |||APSS↓|–|–|*2.25*|1.68|1.05|0.79|
> |||CoverageEfficiency(%)↑|–|–|*23.2*|25.4|35.2|36.2|
> |CPR|PQL|ECR(%)↑|*71.3*|*62.7*|*53.1*|*45.2*|*39.0*|*38.5*|
> |||APSS↓|*20.85*|*5.87*|2.65|*1.51*|*1.09*|*0.95*|
> |||CoverageEfficiency(%)↑|*6.57*|*10.7*|20.0|*29.9*|*35.8*|*40.5*|
>
> “–” indicates failure to satisfy the coverage guarantee (ECR < $1 - \alpha$)
>
> **W1: Writing and Organizations**
>
> We thank the reviewer for the feedback on our narrative flow. As our work intersects CP and KGQA, the introduction follows a three-step progression: (1) establishing the need for framing trustworthy KGQA as a set-valued prediction problem;(2) identifying standard CP’s multi-hop reasoning failures from sequential dependencies and large set sizes. These serve as motivations for (3) introducing our CPR framework as a principled solution to address the identified gaps. We will strengthen logical transitions in the revision.
>
> To enhance the clarity, we will integrate Figure 1 to the introduction as a structural roadmap for introducing the methodology and core condepts in the revision. The three panels in Figure 1 will be referenced with their corresponding sections:
> * **Left Panel** illustrates the pitfalls of hop-level calibration (Section 3.3).
> * **Middle Panel** details the discriminative scoring via RCVNet and PUCT (Section 4.1).
> * **Right Panel** introduces the Query-Exchangeable Conformal Prediction (Section 4.2).

---

> > ### Author Rebuttal · Reviewer_vo6x · 2026-04-03
> >
> > Thanks for the authors' rebuttal. I will increase the score accordingly.

---

> > > ### Author Response · Authors · 2026-04-03
> > >
> > > We sincerely thank the reviewer for taking the time to review our rebuttal and for recognizing the contributions of our work.
> > >
> > > We are encouraged that our rebuttal has convincingly addressed your concerns. We will incorporate these clarifications into the final version, further sharpen the presentation of our technical contribution, and refine the overall quality of writing. Thank you again for your constructive feedback!

---

### Decision · Program_Chairs · 2026-04-30

**Decision:**

Accept (regular)

**Comment:**

The paper explores robustness and calibration of KGQA methods through the lens of conformal prediction and derives conformal path reasoning (CPR) to estimate the uncertainty of reasoning paths via MCTS. Reviewers highlight the importance of the problem as well as novelty and soundness of the proposed method. Experiments are conducted against most recent baselines which is commendable. The authors did a good job addressing reviewers' concerns during the rebuttal running evals on more datasets, estimating reliance on LLM predictions, and estimating scalability of the method. Therefore, I am confident to recommend the acceptance.